



# Evaluation of the Vertical Distribution of Particle Shape (VDPS) method with in situ measurements and assessment of the impact of non-Rayleigh scattering

Audrey Teisseire[1], Patric Seifert[1], Kevin Ohneiser[1], Maximilian Maahn[2], Robert Spirig[3], and Jan Henneberger[3]

[1]Leibniz Institute for Tropospheric Research, Leipzig, Germany
[2]Leipzig Institute for Meteorologie (LIM), Leipzig University, Leipzig, Germany
[3]Institute for Atmospheric and Climate Science, ETH Zurich, Zurich, Switzerland

**Correspondence:** Audrey Teisseire: teisseire@tropos.de

**Abstract.**

In this study, the vertical distribution of particle shape (VDPS) method for retrieval of the vertical distribution of particle shapes and the identification of riming and aggregation processes is evaluated through comparison with in-situ measurements and co-located multi-frequency radar observations collected during the CLOUDLAB campaign in Eriswil, Switzerland. Addi-
tionally, a novel aspect of the VDPS method is introduced, enabling the derivation of the polarizability ratio using slanted-mode linear depolarization ratio (SLDR) calculated from the main peaks of the Doppler spectra of the signal-to-noise ratio (SNR) from the co- and cross-polarized channels, respectively. This enhancement allows for the detection of secondary ice production and the coexistence of multiple hydrometeor types, which would stay undetected when the retrieval is only applied to the main peak of the Doppler spectrum in the co channel. Finally, the susceptibility of the VDPS method to the effects of non-Rayleigh
scattering (particle sizes close or larger than the radar wavelength) is examined. The obtained results were found to demonstrate the potential of the VDPS method using a Ka-band scanning cloud radar in SLDR mode for operational hydrometeor classification, even under non-Rayleigh scattering conditions.

## 1 Introduction

Understanding the processes governing ice-phase precipitation is critical, as more than $60\%$ of global surface precipitation
originates from ice-phase mechanisms (Mülmenstädt et al., 2015; Heymsfield et al., 2020). While the physics of vapor-grown ice crystals such as plates, dendrites, and columns is relatively well established (Lamb and Scott, 1974; Libbrecht, 2003), more complex microphysical processes like riming, graupel and hail formation, and aggregation remain less well understood (Jiang et al., 2019). Crystal growth is strongly influenced by temperature and supersaturation, which govern the resulting hydrometeor shapes or habits (Pruppacher et al., 1998; Libbrecht, 2005). Depending on temperature and humidity conditions
between $-40°C$ and $0°C$, ice crystals can form plate-like or columnar structures (Bailey and Hallett, 2009). In particular, mixed-phase clouds, typically occurring in this temperature range, play a key role in these microphysical transformations.



Riming and aggregation processes influence strongly the precipitation rate by modifying the size, density and fall velocity of ice particles. Riming and aggregation are two fundamental microphysical processes that govern the growth and evolution of ice particles in mixed-phase and cold clouds. Riming occurs when supercooled liquid droplets collide with and freeze onto

falling ice crystals, leading to the formation of denser, more massive particles such as graupel and hail (Pruppacher and Klett, 1996; Lamb and Verlinde, 2011). This process enhances particle mass and fall speed, playing a key role in rapid precipitation formation (Kneifel et al., 2015). Rimed particles in early stages preserve elements of their initial crystalline structure but become increasingly coated with supercooled droplets, leading to compact shapes. In contrast, graupel particles formed through intense riming, are nearly spherical, densely structured, and fall at relatively high terminal velocities of typically between

$v = -1.5$ and $v = -2\,\mathrm{m\,s^{-1}}$ (Kneifel et al., 2016; Kneifel and Moisseev, 2020; Vogl et al., 2022) whereas pristine crystals fall more slowly with velocities of $v > -1\,\mathrm{m\,s^{-1}}$. In contrast, aggregation refers to the process of agglomeration of ice crystals during collisions, producing for example larger, low-density snowflakes composed of multiple interlocked crystals. While riming contributes to compact and fast-falling particles, aggregation results in fluffy, slower-falling hydrometeors (Moisseev et al., 2015). Being able to distinguish between these growth mechanisms is critical for improving quantitative precipitation

estimation and refining microphysical parametrizations in weather and climate models (Chellini and Kneifel, 2024).

Recent studies have focused on detecting and distinguishing riming from aggregation processes using a 35 GHz scanning cloud radar (MIRA-35), by measuring the polarimetric parameter slanted linear depolarization ratio (SLDR) alongside spectral analysis techniques (Teisseire et al., 2025). As a first step, the Vertical Distribution of Particle Shape (VDPS, Teisseire et al., 2024) method is applied to retrieve the vertical profile of particle shapes. Isometric particles observed at temperatures up to

$0°C$ are then classified as graupel or aggregates. No distinction can be made between these two categories at this stage. In a subsequent step, the isometric particle species is classified as graupel or aggregates based on several microphysical parameters, such as the temperature range and the presence or absence of supercooled liquid droplets. However, SLDR can be affected by the presence of large or dense particles causing non-Rayleigh scattering effects to become significant. Indeed, Non-Rayleigh scattering occurs when the target particles (e.g., large raindrops, hail, graupel, or ice aggregates) are comparable or larger

in size with respect to the radar wavelength, causing complex scattering behavior that deviates from the simpler Rayleigh approximation. This results in enhanced backscatter and altered polarimetric signatures, requiring full Mie or T-matrix theory for accurate interpretation (Lamer et al., 2021). Nonetheless, recent studies suggest that the influence of non-Rayleigh scattering on polarimetric variables such as (S)LDR remains relatively modest (Matrosov, 2021).

This study aims to evaluate the capabilities of the VDPS method in characterizing the vertical distribution of particle shape

(Teisseire et al., 2024) as well as of its applicability of the detection of riming and aggregation processes (Teisseire et al., 2025). Prerequisite for the approach is the extensive set of in-situ and multi-wavelength cloud radar observations that was collected in winter 2023/2024 during the CLOUDLAB campaign at Eriswil, Switzerland. The paper is structured as follows. Section 2 gives an overview of the field campaign and the instrumentation. In Section 3 the methods used in this article are described. Section 4 is divided in two parts. The first part is dedicated to the comparison of the VDPS results with in-situ measurements.

Two case studies featuring large aggregates and dense graupel are presented in the second part, in order to evaluate the possible




effect of non-Rayleigh scattering on SLDR measurements and the corresponding influence on the particle shapes derived by the VDPS method.

## 2  Datasets

### 2.1  CLOUDLAB campaign

With the goal to deepen the understanding of cloud physics, a collaboration of the Swiss Federal Institute of Technology (ETH) Zurich and Leibniz Institute for Tropospheric Research (TROPOS) conducted targeted seeding experiments in Eriswil, Switzerland (47.071°N, 7.874°E, 920 m  above sea level, a.s.l.). The setup of CLOUDLAB is described in detail in Henneberger et al. (2023). The CLOUDLAB experiments focus on studying cloud processes and precipitation formation, utilizing a comprehensive range of in-situ and remote-sensing observations and numerical modeling techniques. During the winter 2023/2024,

the CLOUDLAB project involved three additional collaborations, two of which were part of the German Science Foundation Priority Program PROM (Polarimetric Radar Observations Meet Modelling, Trömel et al. (2021)). The project PolarCAP (Polarimetric observations of Clouds and Precipitation) involved a deployment of the entire mobile platform LACROS (Leipzig Aerosol and Clouds Remote Observations System, Radenz et al., 2021; Ohneiser et al., 2025) to the CLOUDLAB field site. The second collaboration, the CORSIPP project (Characterization of Orography-Influenced Riming and Secondary Ice Produc-

tion and Their Effects on Precipitation Rates Using Radar Polarimetry and Doppler Spectra), is led by the Leipzig Institute of Meteorology (LIM) of University of Leipzig and utilized a 94 GHz cloud radar and the Video in Situ Snowfall Sensor (VISSS). The objective of this project is to investigate the microphysical processes that lead to the formation of distinct hydrometeor populations. Finally, the third collaboration was conducted with École Polytechnique Fédérale de Lausanne (EPFL), which employed an X-band radar and MASC (Multiple Angle Snowflake Camera) for in-situ measurements of ice crystal properties.

As a result, the CLOUDLAB campaign 2023/2024 became one of the largest joint deployments of multi-wavelength radar and lidar systems ever. Further details on the instrumentation are provided in the following subsections.

### 2.2  Instrumentation

The primary subset of instruments from the CLOUDLAB winter campaign 2023/2024 that was utilized in the framework of this study is listed in Table 1. This study involves three cloud radars: one modified and one standard MIRA-35 operating at Ka-

band, and one cloud radar of type RPG-94-FMCW operating at W-band. The full set of instrumentation during the 2023/2024 CLOUDLAB campaign was depicted by Ohneiser et al. (2025).

Firstly, MIRA-35 MBR5 is a scanning 35 GHz Doppler cloud radar, a modified version of the conventional LDR mode cloud radar, operated in SLDR mode. This device was part of the CLOUDLAB equipment of ETH Zurich. Unlike the standard LDR mode, SLDR measurements are less influenced by variations in hydrometeor orientation, even at low elevation angles

(Matrosov, 1991; Matrosov et al., 2001). In this configuration, it transmits slant linear polarized pulses and receives in both co- and cross-polarized channels, enabling the detection of particle shape and orientation, particularly useful for distinguishing



between ice crystal habits and liquid droplets. SLDR measurements from the 35-GHz cloud radar MIRA-35 MBR5, operating in SLDR mode (Reinking et al., 2002; Matrosov et al., 2012), serve as the foundation for the VDPS method applied in this study (Teisseire et al., 2024, 2025). The co-cross correlation coefficient ($\rho_{cx}$) is also collected by MIRA-35 MBR5. It can be used as an additional constraint in the determination of the particle orientation (Myagkov et al., 2016b)). As part of this study, MIRA-35 MBR5 conducted range height indicator (RHI) scans, sweeping from $90°$ (zenith) to $150°$ elevation (equivalent to $30°$ elevation angle), at an angular velocity of $0.5°\,\mathrm{s}^{-1}$ . These RHI scans enable one to capture the polarimetric signatures of hydrometeors within the observation volume. The specified elevation angle notation is used consistently throughout the article. A detailed description of the standard LDR mode MIRA-35 was provided in Görsdorf et al. (2015).

Another 35 GHz cloud radar of the LACROS suite operated in Simultaneous Transmit Simultaneous Receive (STSR) scanning mode (MIRA-35 MBR7) and was co-located with MIRA-35 MBR5. This cloud radar is used in this study to provide zenith-pointing LDR overviews, which are comparable to SLDR measurements in the zenith direction.

The third from the row of cloud radars utilized for this study is the zenith-pointing, dual-polarization radar RPG94 of type (RPG-FMCW-94-DP), which is operating at 94 GHz and was manufactured by Radiometer Physics GmbH. It employs Frequency-Modulated Continuous Wave (FMCW) signals to provide vertically resolved profiles of the Doppler spectra of SNR in the co and cross channel, radar reflectivity factor $Z_\mathrm{e}$, and LDR. RPG94 serves two main purposes in this study. First, it is used together with data from MIRA-35 in the computation of the Dual-Wavelength Ratio ($\mathrm{DWR_{Ka-W}}$) and to provide continuous vertical-stare measurements of $Z_\mathrm{e}$ and LDR when scanning was performed by both MIRA-35 MBR5 and MBR7 cloud radars (see Tab. 1).

The Video In Situ Snowfall Sensor (VISSS) was co-located beside the cloud radars during the CLOUDLAB campaign and characterizes particle shape and size of snowfall hydrometeors $1.5\,\mathrm{m}$ above the ground. The VISSS system is composed of two cameras equipped with LED backlights and telecentric lenses, enabling precise particle sizing. Its design combines a large observation volume with high pixel resolution while minimizing wind-induced disturbances. VISSS data products include a range of particle characteristics such as maximum dimension, cross-sectional area, perimeter, shape complexity, and fall velocity. Maahn et al. (2024) demonstrates that VISSS delivers reliable statistics, capturing up to $10000$ distinct particle observations per minute . This instrument is used as an in situ measurement reference in this study to compare the hydrometeors observed at the surface with the particle shapes retrieved by the VDPS method.

## 3 Methods

### 3.1 VDPS method

The VDPS method is detailed in Teisseire et al. (2024), where it is validated through three case studies representing the three main particle shape categories: prolate, isometric, and oblate. The VDPS method used a SLDR mode scanning cloud radar (MIRA-35 MBR5 in this study) and is designed to characterize the shape of cloud particles using. The employed spheroidal scattering model relates the elevation-angle dependency of SLDR with the polarizability ratio ($\xi$) and the degree of orientation ($\kappa$). $\xi$ and $\kappa$ describe the apparent particle shape through a density-weighted axis ratio and their preferred orientation, respec-



**Table 1.** Technical characteristics of the instruments described in Sec. 2.1 during the deployment of the CLOUDLAB campaign in Erwisil, Switzerland. The table lists only the parameters utilized in this study, although the instruments are able of providing additional parameters.

| Data source (Reference) | Frequency $\nu$ Wavelength $\lambda$ | Measured / retrieved quantity | Temporal resolution | Range | Range resolution |
|---|---|---|---|---|---|
| **SLDR-mode scanning cloud radar MIRA-35 MBR5** Metek company MIRA-35-SLDR (Görsdorf et al., 2015; Teisseire et al., 2024) | $\nu = 35.2\,\text{GHz}$ | Signal-to-noise ratio SNR Slanted linear depolarization ratio SLDR Cross correlation coefficient $\rho_{cx}$ Radar reflectivity factor $Z_e$ | 1 s | 150 – 15000 | 31.18 m |
| **STSR-mode scanning cloud radar MIRA-35 MBR7** Metek company MIRA-35-STSR (Myagkov et al., 2015) | $\nu = 35.2\,\text{GHz}$ | Signal-to-noise ratio SNR Differential reflectivity ZDR Radar reflectivity factor $Z_e$ Linear depolarisation ratio LDR | 1 s | 150 – 15000 | 31.18 m |
| **Doppler cloud radar RPG94** RPG-FMCW-94-DP (Küchler et al., 2017) | $\nu = 94\,\text{GHz}$ | Spectral reflectivity $sZ_e$ Radar reflectivity factor $Z_e$ Mean Doppler velocity $\bar{v}_D$ Linear depolarization ratio LDR | 5 s | 120 – 12000 m | 30 – 45 m |
| **VISSS Video In Situ Snowfall Sensor** (Maahn et al., 2024) | $\nu = 140\,\text{Hz}$ | Particle number concentration Hydrometeor type, shape and size Oblateness Aspect ratio | 60 s | 1.5 | - |
| **Weather model forecast** ECMWF IFS (Owens and Hewson, 2018) | - | Temperature $T$ Pressure $P$ Relative Humidity RH | 3600 s | 10 – 12000 m | 20 – 300 m |

tively (Myagkov et al., 2016a; Teisseire et al., 2024). A polarizability ratio $\xi \approx 1$ (when $\xi$ ranges from 0.8 to 1.2, depending on the radar calibration) corresponds to isometric particles, representing spherical particles or those with low density which appear as isometric ones to the radar. In contrast, a polarizability ratio $\xi < 0.8$ or $\xi > 1.2$ describes oblate and prolate particles, respectively. The utilization of the VDPS method in combination with auxiliary measurement techniques was demonstrated to capture the occurrence of riming and aggregation processes (Teisseire et al., 2025). In the current version of the method, SLDR

is calculated based on the main peak of the Doppler spectrum of the SNR measured in the co-polarized channel ($\text{SNR}_{\text{co}}$), which represents the dominant hydrometeor population with the highest backscattering cross-section. An alternative approach proposed in this paper calculates SLDR using the main peak of the Doppler spectrum of SNR measured in the cross-polarized channel ($\text{SNR}_{\text{cross}}$) highlighting hydrometeor populations that induce significant depolarization, such as columnar ice crystals. By comparing the polarizability ratio $\xi$ derived by the VDPS method using the main peak signatures from both $\text{SNR}_{\text{co}}$ and

$\text{SNR}_{\text{cross}}$, respectively, it is possible to identify microphysical processes that generate distinct hydrometeor populations, such as secondary ice production (SIP), as will be demonstrated in Sec. 4.3.





## 3.2 Dual wavelength ratio (DWR)

The difference between the measured reflectivity within a cloud at 35 GHz and 94 GHz can be used to detect large ice particles. Indeed, high values of $\mathrm{DWR_{Ka-W}}$ suggest the departure of the W-band radar from the Rayleigh scattering regime, indicating

the presence of large aggregates or dense graupel, whereas values of $\mathrm{DWR_{Ka-W}} \approx 0$ correspond to small hydrometeors. In order to compute $\mathrm{DWR_{Ka-W}}$ the following steps are performed. First, the observed reflectivity $Z_e$ from MIRA-35 MBR7 and RPG94 is mapped onto a common time-height grid using nearest-neighbour interpolation. The attenuation due to atmospheric gases, including water vapour, is estimated using the Passive and Active Microwave TRAnsfer (PAMTRA) tool (Mech et al., 2020) and based on the temperature and humidity profiles from the European Centre for Medium-Range Weather Forecasts

Integrated Forecasts System (ECMWF-IFS). Since $\mathrm{DWR_{Ka-W}}$ is only used qualitatively in this study, no correction for liquid water attenuation is applied. MIRA-35 MBR7 reflectivity $Z_e(\mathrm{Ka})$ is offset-corrected to align with RPG94 reflectivity $Z_e(\mathrm{W})$ in the Rayleigh regime, following the approach outlined by Dias Neto et al. (2019). This correction is performed using 30-minute time intervals, selecting cloud regions that are at least 1 km above the $0°\mathrm{C}$ isotherm and 4 km above the surface, where $Z_e(\mathrm{Ka})$ ranges from $-30$ to $-10\,\mathrm{dBZ}$. In these regions, both cloud radars are expected to operate within the Rayleigh regime.

The reflectivity difference, $Z_e(\mathrm{W}) - Z_e(\mathrm{Ka})$, represents the offset, which is then applied to $Z_e(\mathrm{Ka})$ for the corresponding 30-minute period. To ensure the quality of the offset correction, the correlation between $Z_e(\mathrm{Ka})$ and $Z_e(\mathrm{W})$ must exceed 0.9 for the selected cloudy pixels.

## 3.3 VISSS processing

The interpretation of hydrometeor shape with VISSS is detailed in Maahn et al. (2024) and follows a multi-step procedure.

First, VISSS detects and analyses hydrometeors in video frames by first identifying motion between intermittent frames, reducing data volume. Particles are identified by background subtraction and the individual particle shapes are determined through dilation, filling and erosion. The final mask is used to compute particle properties like maximum size, area, perimeter, aspect ratio, canting angle, and complexity. Some quality criteria are applied: only particles with the maximal diameter $D_{\max} \geq 2$ pixels are retained. The VISSS combines particles detected by two separate cameras into a single 3D coordinate system, dis-

carding those only visible in one camera. This is obtained by comparing the vertical position and extent of particles, assuming normally distributed measurement errors. The tracking of matched particles over time with VISSS enables the reconstruction of their 3D trajectories, allowing estimation of sedimentation velocity and interaction with turbulence. Natural tumbling provides multiple perspectives, improving estimates of particle properties such as the diameter, area, perimeter, and aspect ratio of particles.

## 4 Results


This section focuses on validating the VDPS method described in Teisseire et al. (2024) and Teisseire et al. (2025), by comparing its results with in situ measurements and spectral retrieval techniques. VISSS observations (see Sec. 3.3) and Doppler




spectrograms are used to compare and validate the results obtained from the VDPS method. In this section the hydrometeor
layers and microphysical processes are described from cloud top to cloud base. A first case study presented in subsection 4.1,
demonstrates the capability of the VDPS method to classify liquid droplets from rain as isometric particles while two other
case studies are shown to link different ice crystal shapes to the underlying microphysical processes. In the second subsection
4.2, the observation of an event depicting the formation of rimed particles is presented, where the riming process is associ-
ated with the observation of dendrites, graupel and rimed dendrites. Finally, in the third subsection 4.3, the Hallett-Mossop
process is highlighted, leading to the formation of rimed aggregates and columnar crystals. To differentiate between these two
hydrometeor formations, the VDPS method utilizes SLDR values derived for the main peak in SNR measurements in the co-
and cross-polarized channels, respectively. This approach aims to distinguish the dominant ice crystal population, in this case
rimed aggregates, from the subset of columnar crystals that exhibit the highest depolarization (see Sec. 3.1). For this study, the
0-m height level corresponds to the Eriswil station altitude, located at $920\,\mathrm{m}$ a.s.l.

## 4.1 Observation of liquid droplets on 22 February 2024, 12:00 UTC

The focus of this first case study is the characterization of liquid precipitation (rain) that was observed at Eriswil on 22 February
2024 from 11:00 to 13:00 UTC. A general overview of the precipitating cloud system is shown in Fig. 1. The black rectangle in
Fig. 1a marks the time period of the RHI scan performed by MIRA-35 MBR5. The radar reflectivity factor $Z_e$ and LDR from
RPG94 are shown in Figs. 1a and 1c, respectively. In Fig. 1b, the $\mathrm{DWR_{Ka-W}}$ calculated with difference between $Z_e(\mathrm{W})$ from
RPG94 and $Z_e(\mathrm{Ka})$ from MIRA-35 MBR7, describes the size of particles.

The melting layer is clearly indicated by a sudden increase in $Z_e$ and LDR at around $1\,\mathrm{km}$ height, as shown in Figs. 1a
and 1c, respectively. The location of the melting layer is confirmed by means of the RHI scan of SLDR by MIRA-35 MBR5
shown in Fig. 2a, where SLDR$\approx -10\,\mathrm{dB}$. The vertical distribution of the polarizability ratio derived by the VDPS method is
presented in Fig. 2b. Between $1.1$ and $0.9\,\mathrm{km}$ above the ground, the VDPS method detects a transition from prolate ($\xi > 1$)
to oblate ($\xi < 1$) shapes which is associated to the melting layer detected previously. Below the melting layer, low values of
SLDR of around $-30\,\mathrm{dB}$ are measured with both MIRA-35 MBR7, and RPG94 as shown in Fig. 2a and 1c, respectively. The
VISSS measurements, presented in Fig. 2c for the time period from 11:14 to 12:31 UTC provide in situ information on the
shape and size of hydrometeors. In Fig. 2c, liquid droplets are detected as spherical particles with the VISSS and described as
isometric particles by the VDPS method which derives a polarizability ratio $\xi \approx 1$ from the melting layer to the ground level,
as shown in Fig. 2b. The comparison between particle shapes derived from the VDPS method and in situ VISSS measurements
demonstrates the ability of the VDPS method to identify liquid droplets as isometric particles, as indicated by a polarizability
ratio of approximately $\xi \approx 1$ from the melting layer down to the ground level.

## 4.2 Rimed dendrites observed on 8 January 2024, 10:00 UTC

The focus in this case study is put on a low-level stratus cloud layer that was observed at Eriswil on 8 January 2024 from
09:00 to 11:00 UTC. An overview on the observation is presented in Fig. 3. During the whole period, a low-level stratus cloud
was located below $1.2\,\mathrm{km}$ height, capped by an inversion layer where temperatures where approximately $-10°\mathrm{C}$ (Fig. 3).



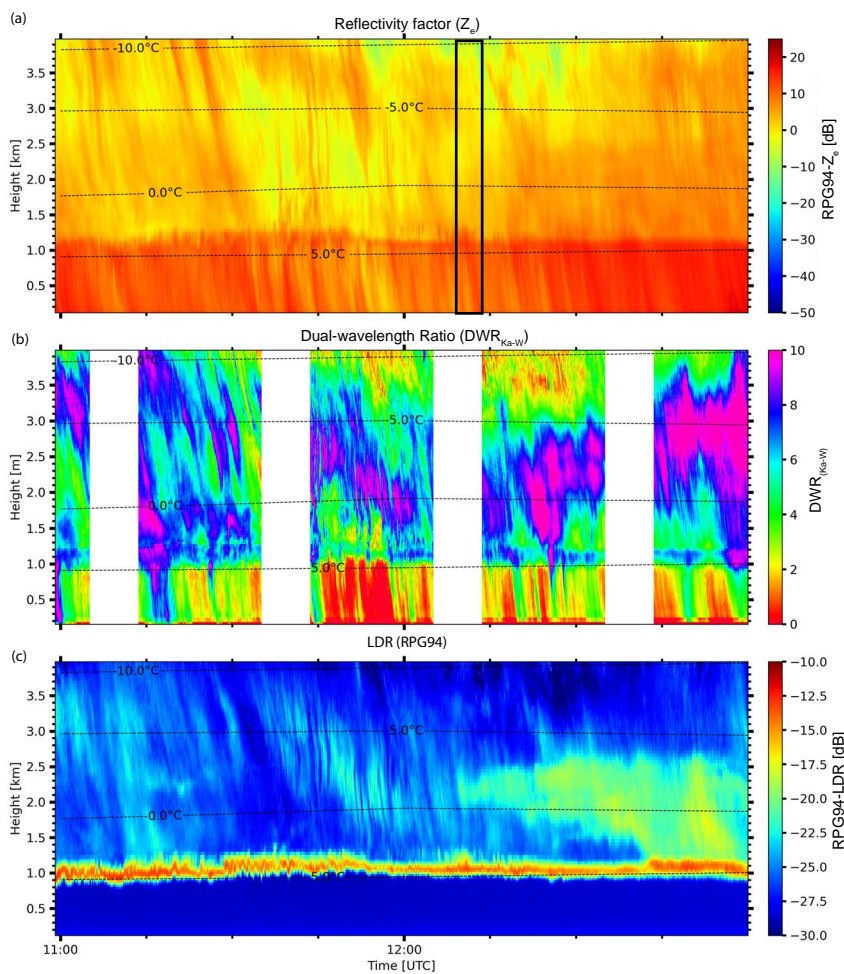

**Figure 1.** Case study of a deep mixed-phase cloud event observed with multi-wavelength polarimetric cloud radars at Eriswil, Switzerland, on 22 February 2024 from 11:00 UTC to 13:00 UTC. (a) Vertically pointing W-band (94 GHz) radar reflectivity factor $Z_e$, (b) represents the dual-wavelength ratio (DWR$_{Ka-W}$, ratio of $Z_e$ between Ka-band MIRA-35 MBR7 and W-band LIMRAD94) and (c) the linear depolarization ratio (LDR) using RPG94. No measurements are available for the time periods where white bands are shown in (b), due to RHI and PPI scans, that were conducted during these periods.





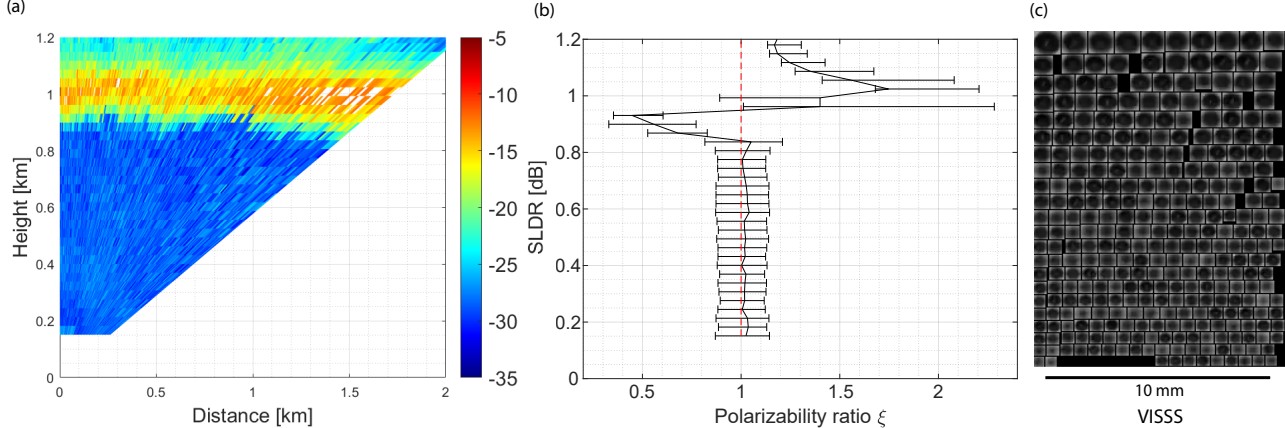

**Figure 2.** (a) RHI scans of SLDR from $90°$ to $150°$ elevation angle recorded by MIRA-35 MBR5 on 22 February 2024, from 12:08:16 to 12:10:17 UTC in Erwisil, Switzerland. (b) shows the vertical distribution of the polarizability ratio $\xi$ derived using the VDPS method and corresponds to the RHI scan presented in (a). (c) The surface observations of the randomly selected hydrometeor shapes with the VISSS from 11:14 to 12:31 UTC during the radar scan period.

The cloud top exhibits low values of $Z_e$ and LDR, as shown in Figs. 3a and 3c, respectively, suggesting the presence of a supercooled liquid layer. Simultaneously, rimed dendrites are visually observed by eye at the site from 10:00 to 11:00 UTC, which is corroborated by the VISSS observations (Fig. 4c). These observations suggest the presence of a dendritic growth layer within the cloud, which would be the prerequisite for any subsequent riming to occur. Two case studies are presented in this
subsection: the case outlined by the blue rectangle at around 10:08 UTC shown in Fig. 3a, which illustrates an early stage of the riming process, resulting in the formation of rimed dendrites, whereas the case highlighted by the red rectangle at around 10:38 UTC represents a more advanced stage of riming, leading to the development of graupel. The color codes defining the boxes are consistently used in Figs. 4 and 5.

As a starting point, the low-LDR layer that is present in both case studies depicted previously in Fig. 3c is examined.
Regarding the Doppler spectrogram depicted in Figs. 5a and 5d, a layer with low values of $SNR_{co}$ associated with a fall velocity $v \approx 0~\mathrm{m\,s^{-1}}$ is observed from 1.2 to 1 km height. The absence of detectable signals in $SNR_{cross}$ and SLDR at this height, as shown in Figs. 5b and 5c, respectively, indicates that these particles are likely supercooled liquid droplets at the top of the cloud. Between 1 and 0.8 km height, low values of SLDR are measured by MIRA-35 MBR5 as shown in Figs. 4a and 4d at all elevation angles. This corresponds to the presence of isometric particles as demonstrated by means of the VDPS
retrievals shown in Figs. 4b and 4e, where $\xi \approx 1$. In parallel, an increase of $SNR_{co}$ is observable in Fig. 5a and low values of $SNR_{cross}$ and SLDR are shown in Figs. 5b and 5c, respectively, with a Doppler velocity $v \approx 0~\mathrm{m\,s^{-1}}$. These features can be assigned to isometrically shaped particles, such as drizzle droplets. Indeed, the presence of an inversion layer can contribute to the sustainment of supercooled liquid water in clouds at temperatures as low as $-10°C$ (Ricaud et al., 2024). Below 0.8 km

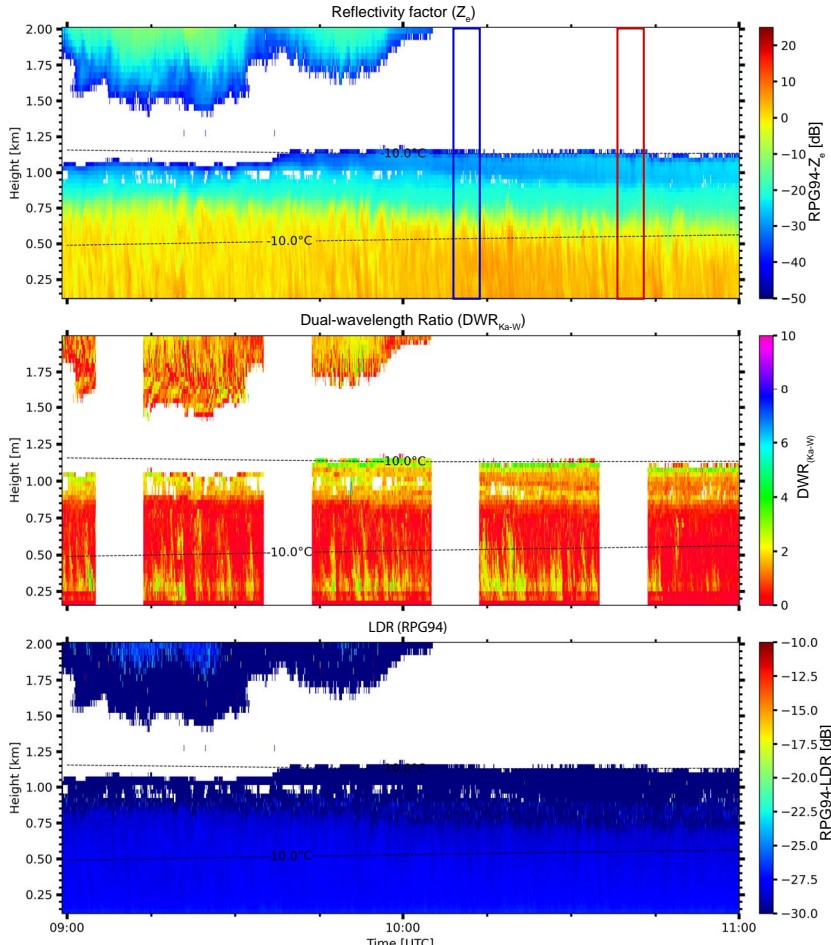

**Figure 3.** Case study of a stratus cloud event observed with multi-wavelength polarimetric cloud radars at Eriswil, Switzerland, on 8 January 2024 from 9:00 UTC to 11:00 UTC. (a) Vertically pointing W-band (94 GHz) radar reflectivity factor $Z_e$, (b) represents the dual-wavelength ratio (DWR$_{Ka-W}$, ratio of $Z_e$ between Ka-band MIRA-35 MBR7 and W-band RPG94) and (c) the linear depolarisation ratio (LDR) using RPG94. No measurements are available for the time periods where white bands are shown in (b), due to RHI and PPI scans, that were conducted during these periods. The blue box indicates the RHI scan conducted at 10:08 UTC, while the red box marks the scan at 10:38 UTC. These color codes are consistently used in Figs. 4 and 5.



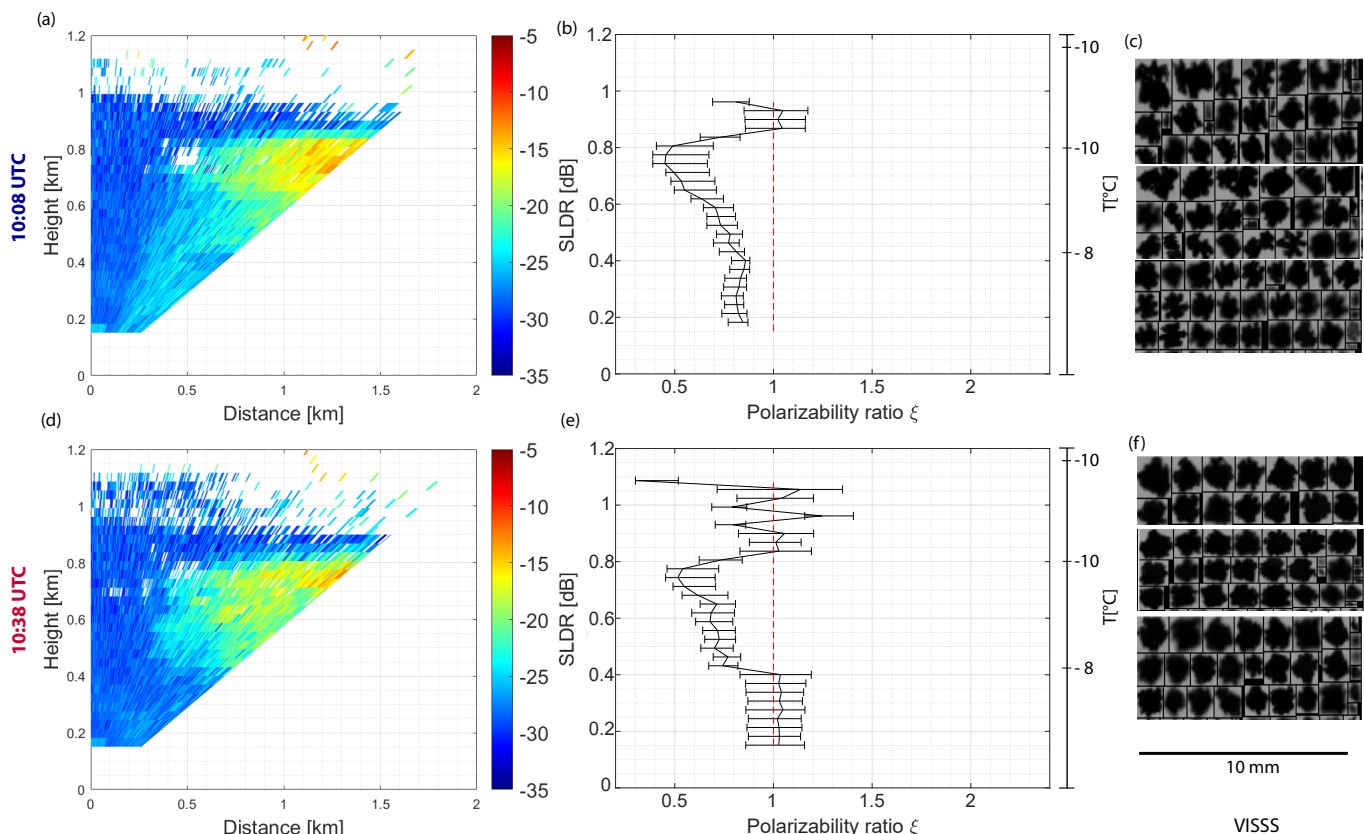

**Figure 4.** RHI scans of SLDR from $90°$ to $150°$ elevation angle recorded by MIRA-35 MBR5 on 8 January 2024, (a) from 10:08:14 to 10:10:17 and (d) from 10:38:14 to 10:40:15 UTC in Erwisil, Switzerland. (b) and (e) show the vertical distribution of the polarizability ratio $\xi$ derived using the VDPS method, with the temperature range on the right side, and corresponding to the RHI scan presented in (a) and (d), respectively. The surface observation of randomly selected hydrometeor shapes is shown in (c) for the time period from 10:15:30 to 10:17:00 and in (f) from 10:45:00 to 10:46:30, corresponding to results presented in (b) and (e), respectively.



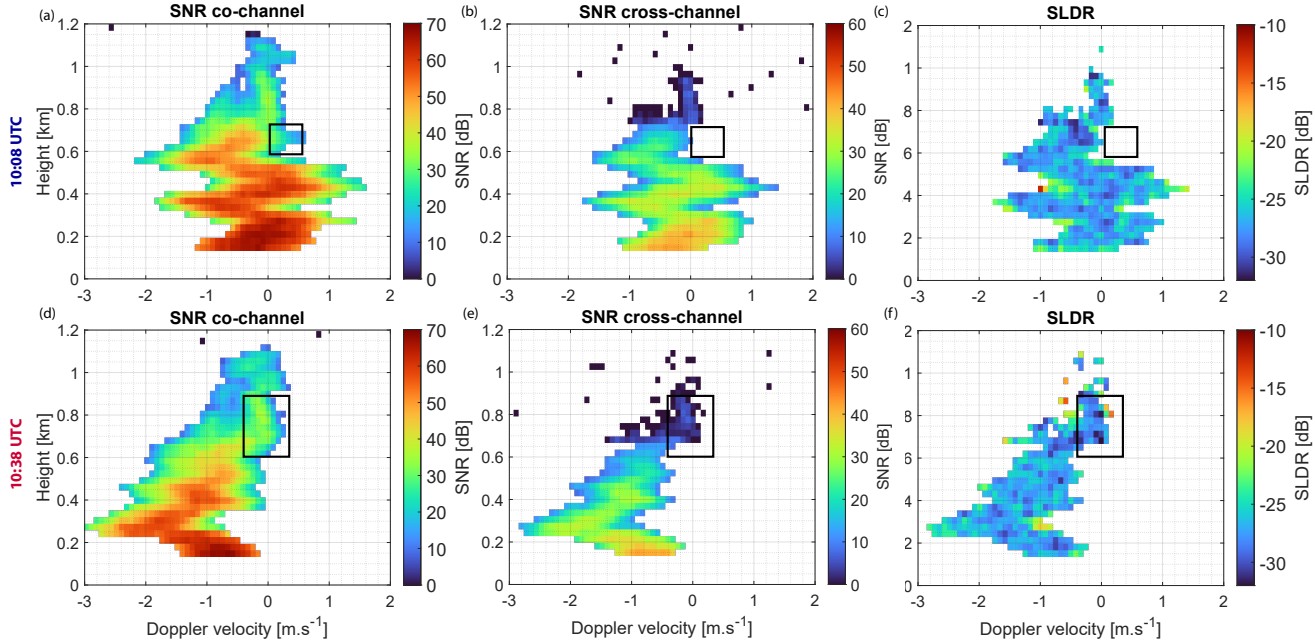

**Figure 5.** Doppler spectrograms at zenith pointing of (a),(d) SNR in the co-channel, (b),(e) SNR in the cross-channel and (c),(f) SLDR measured with MIRA-35 MBR5 on 8 January 2024, at 10:08:16 (upper panels) and 10:38:14 UTC (lower panels), respectively. The black boxes show where the presence of supercooled liquid droplets was identified.

height, another hydrometeor population is present which is characterized by Doppler velocities $v \approx -1\,\mathrm{m\,s^{-1}}$, as shown in

Fig. 5a, while low values of $\mathrm{SNR_{cross}}$ and SLDR are presented in Figs. 5b and 5c, respectively. At this altitude, the VDPS method derives oblate particles according to Figs. 4b and 4e, which are characterized by low values of the polarizability ratio $\xi = 0.5$. At the same altitude, the temperature is approximately $-10°C$, allowing the formation of plate-like crystals.

    Next, the blue-framed case from 10:08 UTC is discussed, that represents an early stage of a riming process. Between 0.6 km height and the ground, the polarizability ratio increases and stabilizes at around $\xi = 0.8$, as shown in Fig. 4b. This indicates

the presence of slightly oblate-shaped particles falling at a velocity not exceeding $v = -1\,\mathrm{m\,s^{-1}}$, as shown in Figs. 5a-c. This phenomenon is attributed to the transformation of ice crystals upon contact with the supercooled liquid droplets that were detected down to 0.6 km height (Figs. 5a and 5d), leading to the transformation of dendritic crystals toward rimed dendrites, particles which appear oblate but more isometric than dendrites, and that fall more slowly than graupel. Regarding the particle shapes presented in Fig. 4c, rimed dendrites are detected using VISSS from 10:15:30 to 10:17:00 UTC. This in-situ observation

confirms the hypothesis that the VDPS-based retrieval of weakly oblate particles, as shown in Fig. 4b below 0.4 km height, is associated to the presence of rimed dendrites.

    Finally, the second red-framed case from 10:38 UTC representing an advanced stage of the riming process is discussed. The polarizability ratio is found to show a decrease at heights below 0.6 km height and stabilizes at $\xi \approx 1$ below 0.4 km height as shown in Fig. 4e. Unlike the previous case, more compact and spherical particles are detected with VISSS from 10:45:00 and





10:46:30 UTC, according to Fig. 4f. In this case, the advanced stage of riming allows the formation of spherical particles such as graupel, which are described by a polarizability ratio $\xi \approx 1$ using the VDPS method. Below $0.4\,\mathrm{km}$ in height, these particles fall faster, reaching velocities of $v \approx -2\,\mathrm{m\,s^{-1}}$, as shown in Figs. 5d-f, due to their higher density.

     In conclusion, the VDPS method identifies the formation of dendrites with a polarizability ratio $\xi = 0.5$ from $0.9$ to $0.7\,\mathrm{km}$ height where temperatures are around $-10°\mathrm{C}$, conditions which are favourable for their development. Close to the surface, in

turn, the presence of rimed dendrites is derived, which is associated to higher polarizability ratios $\xi \approx 0.8$. The riming process alters both the structure and shape of ice crystals, causing dendrites to appear more isometric to the radar. Finally, an advanced stage of riming produces spherical graupel particles, which are characterized by a polarizability ratio $\xi \approx 1$.

### 4.3    Aggregates and columns observed on 6 January 2024

The following analysis is divided into two separate case studies. In Fig. 6 the observation of a convective cloud system is

presented that was observed on 6 January 2024 from 16:00 to 23:00 UTC. The first case study, described in Sec. 4.3.1, is highlighted by a blue rectangle in Fig. 6a and was observed at around 17:40 UTC when aggregates formed. The second case study, described in Sec. 4.3.2, is highlighted by the red rectangle and was observed at around 22:08 UTC when two ice particle types were detected.

### 4.3.1    Observation of aggregates

In Figs. 6a and 6b it is shown, that the first case study is characterized by high values of $Z_{\mathrm{e}}$ measured by RPG94, and associated with high values of $\mathrm{DWR_{Ka-W}}$ from $2.8\,\mathrm{km}$ height to the surface, where the temperature ranges from $-15°\mathrm{C}$ to $0°\mathrm{C}$. As an entry to the interpretation of the first case study, the RHI scan of SLDR and the vertical distribution of the polarizability ratio $\xi$ depicted in Figs. 7a and 7b, respectively, are introduced. At the cloud top located above $3\,\mathrm{km}$ height, low SLDR values are detected at all elevation angles, as shown in Fig. 7a. This population produced a considerable signal to be observable in SLDR

and is characterized by a polarizability ratio of $\xi \approx 1$ (Fig. 7b), highlighting the presence of isometric particles such as small aggregates (Fig. 7b). Regarding vertical profiles of the Doppler spectrograms presented in Fig. 8, low values of $\mathrm{SNR_{co}}$ observed at around $2.8\,\mathrm{km}$ height, as shown in Fig. 8a, are not accompanied by any corresponding $\mathrm{SNR_{cross}}$ or SLDR signals (Figs. 8b and 8c, respectively) and are characterized by a fall velocity of $v \approx 0\,\mathrm{m\,s^{-1}}$. This indicates the presence of supercooled liquid droplets, highlighted in the black box in Fig. 8a. Another hydrometeor population is present at this altitude, represented in the

red rectangle with higher values of $\mathrm{SNR_{co}}$ and a fall velocity of $v \approx -1\,\mathrm{m\,s^{-1}}$.

     Below the supercooled liquid layer, the increasing fall velocity of the isometric hydrometeor population suggests the onset of riming, leading to the formation of denser aggregates. This process results in ice particles exhibiting higher SLDR values at $v \approx 0.8\,\mathrm{m\,s^{-1}}$ which are co-located to the previously identified isometric particle population (Fig. 8c). Below $2\,\mathrm{km}$ height, only prolate particles are derived by the VDPS method, as shown in Fig. 7b. There, temperatures exceeding $-10°\mathrm{C}$ allow for

the growth of columnar crystals. Then, at altitudes below $0.5\,\mathrm{km}$ height, $\mathrm{SNR_{co}}$ increases while lower values of SLDR are observed at all elevation angles of the RHI scan (Figs. 8c and 7a). This low-level hydrometeor population is characterized by isometric particles with a polarizability ratio $\xi \approx 1$, as determined by the VDPS method (Fig. 7b). During the same time period,

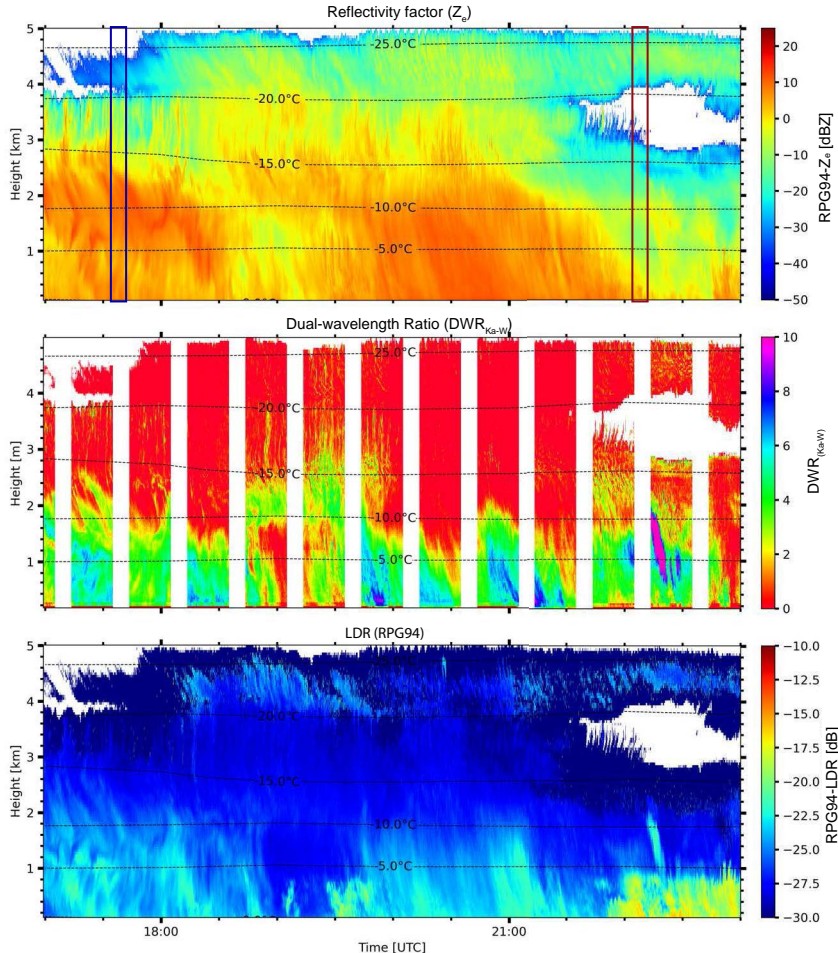

**Figure 6.** Case study of a deep mixed-phase cloud event observed with multi-wavelength polarimetric cloud radars at Eriswil, Switzerland, on 6 January 2024 from 16:00 UTC to 23:00 UTC. (a) Vertically pointing W-band (94 GHz) radar reflectivity factor $Z_e$, (b) represents the dual-wavelength ratio ($DWR_{Ka-W}$, ratio of $Z_e$ between Ka-band MIRA-35 MBR7 and W-band RPG94) and (c) the linear depolarisation ratio (LDR) using RPG94. No measurements are available for the time periods where white bands are shown in (b), due to RHI and PPI scans, that were conducted during these periods. The blue box indicates the RHI scan conducted at 17:38 UTC (discussed in Sec. 4.3.1), while the red box marks the scan at 22:08 UTC (discussed in Sec. 4.3.2).





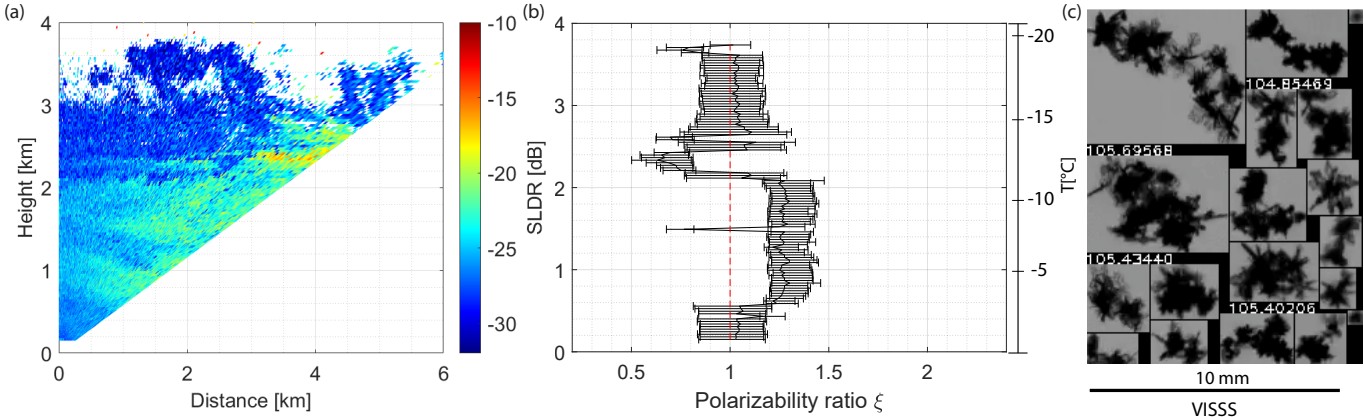

**Figure 7.** (a) RHI scan of SLDR from $90°$ to $150°$ elevation angle recorded by MIRA-35 MBR5 on 6 January 2024, from 17:38:16 to 17:40:16 UTC in Erwisil, Switzerland. (b) Shows the vertical distribution of the polarizability ratio $\xi$ derived using the VDPS method and corresponds to the RHI scan presented in (a), with the temperature range on the right side. The surface observation of randomly selected ice crystal shapes is shown in (c) for the time period from 17:37:01 to 17:58:55 UTC.

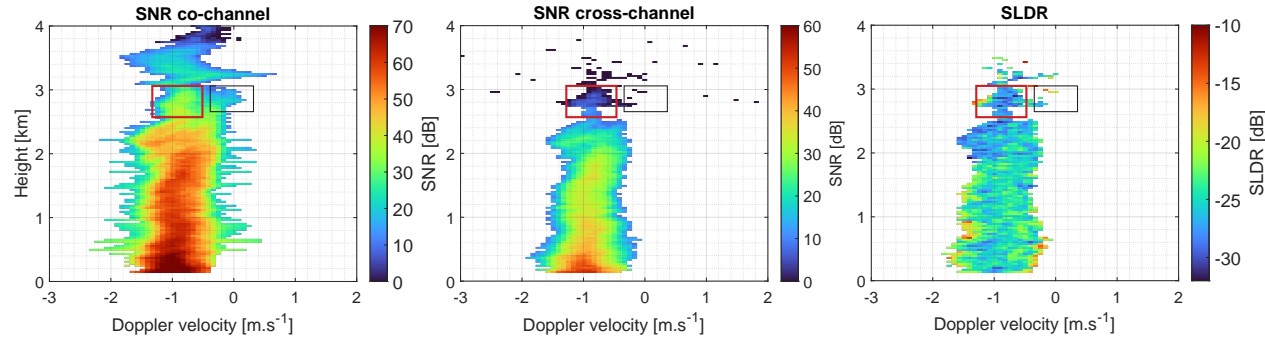

**Figure 8.** Doppler spectrograms at zenith pointing of (a) SNR in the co-channel, (b) SNR in the cross-channel and (c) SLDR measured with MIRA-35 MBR5 on 6 January 2024, at 17:38:16 UTC. The black boxes show the supercooled liquid droplets while the red boxes highlight another co-located hydrometeor population.

the presence of large particles is indicated by the $\mathrm{DWR_{Ka-W}}$ and large aggregates are detected by the VISSS as shown in Fig. 6b and 7c, respectively. The aggregates observed at the surface correspond to a polarizability ratio $\xi \approx 1$ as derived with the VDPS method, demonstrating its capability to identify aggregates as isometric particles.

### 4.3.2 Observation of rimed aggregates and co-located needles

The second case study from the 6 January 2024 observation occurred five hours after the first case study and is dedicated to the detection of the impact of complex microphysical processes leading to secondary ice production. In the time interval





highlighted by the red rectangle in Figs. 6a and 6c, low values of $Z_\mathrm{e}$ are associated to high LDR values below 1 km height,
respectively. This coincides with increasing $\mathrm{DWR_{Ka-W}}$ values, indicating the presence of relatively large particles below
1.5 km height (Fig. 6b).

As an entry to the interpretation of this second case study, the profiles of the Doppler spectra of $\mathrm{SNR_{co}}$, $\mathrm{SNR_{cross}}$ and SLDR
depicted in Fig. 10a–c, respectively, are presented. Regarding the height range from 3 to 2.5 km height in Fig. 10a, $\mathrm{SNR_{co}}$
shows low values in the first branch on the right with a particle fall velocity near $v \approx 0\ \mathrm{m\,s^{-1}}$. These features are not correlated
with any signals of $\mathrm{SNR_{cross}}$ and SLDR in Figs. 10b and 10c, respectively. This observation indicates the presence supercooled
liquid droplets at the cloud top. Another hydrometeor population near cloud top characterized by a particle fall velocity between
$v \approx -0.8\ \mathrm{m\,s^{-1}}$ and $v \approx -1\ \mathrm{m\,s^{-1}}$, as shown in Fig. 10a, suggests the presence of ice crystals such as plate-like ones or small
aggregates, which likely experienced riming by contact with the supercooled liquid, which would explain the increase in fall
velocity.

Secondly, the RHI scans of SLDR and the vertical distribution of the polarizability ratio are presented and correlated with
spectral information. Here, a novel aspect of the VDPS method is introduced. SLDR is calculated from $\mathrm{SNR_{co}}$ and $\mathrm{SNR_{cross}}$
for either the location of the main peak of the Doppler spectrum of $\mathrm{SNR_{co}}$ or $\mathrm{SNR_{cross}}$, respectively. The VDPS retrievals for
both approaches are presented in Fig. 9(a-b) and 9(d-e), respectively. The extended VDPS method allows for a clearer sepa-
ration of the main hydrometeor population (dominating $\mathrm{SNR_{co}}$) from co-located populations of highly depolarizing particles
(dominating $\mathrm{SNR_{cross}}$). Below 1.5 km height, the VDPS method using SLDR calculated from the main peak of the $\mathrm{SNR_{co}}$
derives only isometric particles associated with low values of SLDR, as is shown in Fig. 9b and 9a, respectively. Since the
Doppler spectra profiles do not show signatures of supercooled liquid in this height range, the occurrence of aggregation is
likely. Near the surface, Fig. 9a in addition shows the presence of higher SLDR values. These are associated to an increase
in the polarizability ratio (Fig. 9b) and are attributed to the dominance of columnar crystals from secondary ice production in
this height region. Indeed, the dominant ice population near the surface shifts, with backscattering by the columnar crystals
becoming more pronounced than that from the co-located aggregates. Consequently, SLDR is increasingly representative of the
columnar crystal population rather than the previously dominant isometric particles, as indicated in Fig. 10a, where the black
dots are corresponding to the main peak of $\mathrm{SNR_{co}}$. Near the surface, the black dots are seen to switch between the two ice
particle populations, thereby biasing the SLDR estimation. In comparison, the VDPS method using SLDR calculated for the
location of the main peak in the Doppler spectrum of $\mathrm{SNR_{cross}}$, represented by the black points in Fig. 10b, identifies prolate
particles at heights below 0.8 km (Fig. 9).

In the profile of the spectrogram of $\mathrm{SNR_{co}}$ in Fig. 10a, multiple spectral modes are visible between 2.5 and 2 km height.
Two distinct particle populations are identified based on the signatures shown in $\mathrm{SNR_{co}}$, $\mathrm{SNR_{cross}}$ and SLDR in this height
range. Simultaneously, as shown in Figs. 9b and 9e the VDPS method using $\mathrm{SNR_{co}}$ classifies isometric particles, such as
aggregates, while the VDPS method using $\mathrm{SNR_{cross}}$ detects oblate particles, such as plate-like crystals. From 2 to 1.5 km
height, a secondary spectral mode is measured with low values of $\mathrm{SNR_{co}}$ where particles are falling with a Doppler velocity
$v \approx 0\ \mathrm{m\,s^{-1}}$, as visible in the black rectangle in Fig. 10a. In Figs. 10b and 10c, no signals are detected in $\mathrm{SNR_{cross}}$ and SLDR,
respectively, which indicates that supercooled liquid droplets are observed at this altitude. Between 1.9 and 1.8 km height,



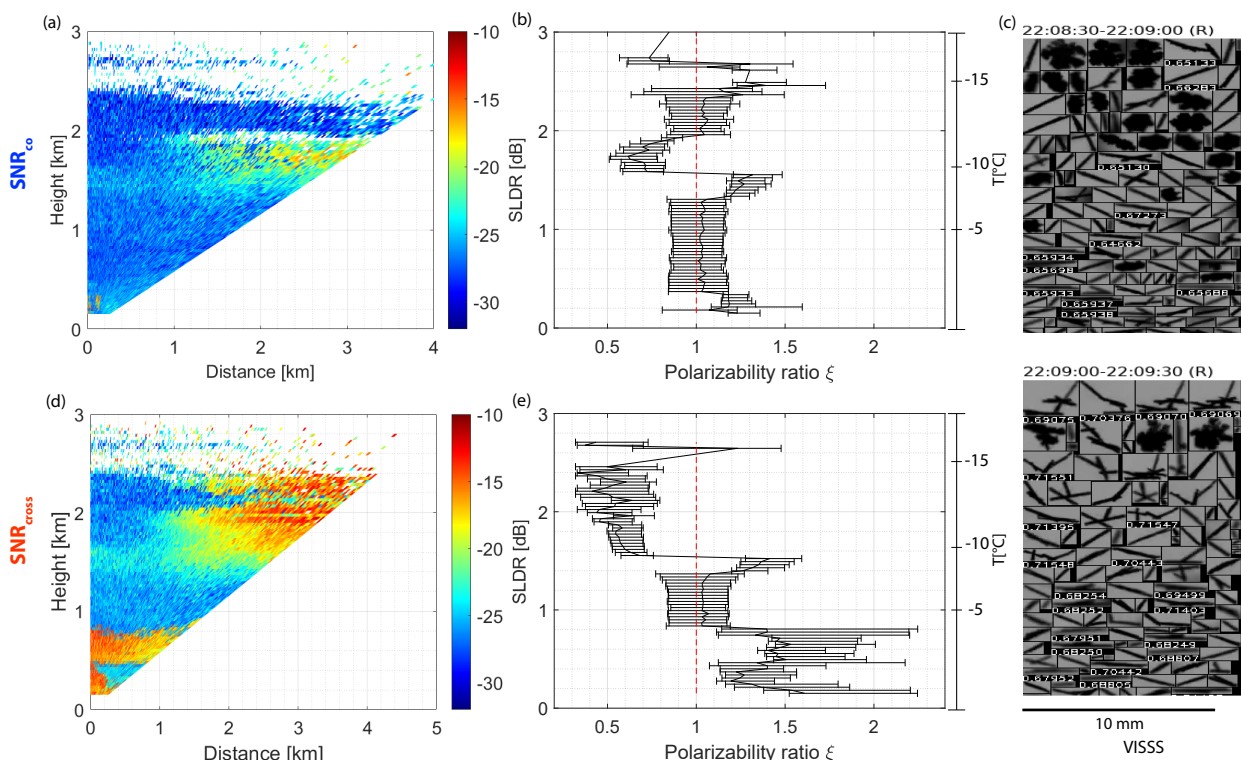

**Figure 9.** (a,d) RHI scan of SLDR from $90°$ to $150°$ elevation angle recorded by MIRA-35 MBR5 on 6 January 2024, from 22:08:06 to 22:10:08 UTC in Erwisil, Switzerland. (b) and (e) show the vertical distribution of the polarizability ratio $\xi$ derived using the VDPS method, as derived from the RHI scans presented in (a) and (d), respectively, with the temperature range on the right side. (a) and (b) are computed based on the main peak of $SNR_{co}$. (d) and (e) are computed based on the main peak of $SNR_{cross}$. The surface observation of randomly selected ice crystal shapes is shown in (c) for the time period from 22:08:30 to 22:09:30 UTC.

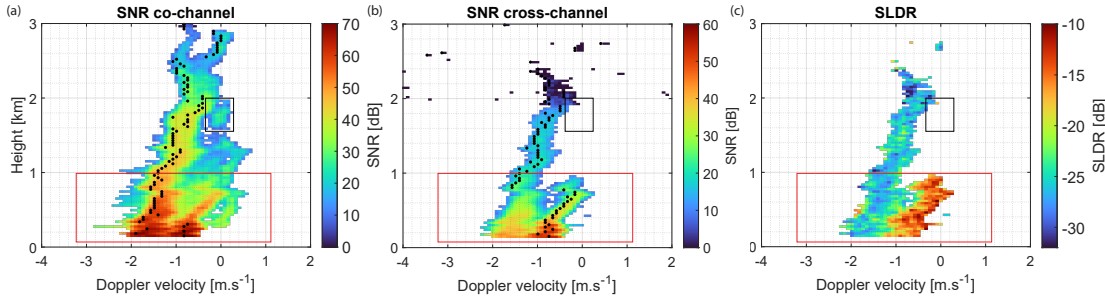

**Figure 10.** Doppler spectrograms at zenith pointing of (a) SNR in the co-channel, (b) SNR in the cross-channel and (c) SLDR measured with MIRA-35 MBR5 on 6 January 2024, at 22:08:16 UTC (last measurement just before the RHI scan). The black points in (a) and (b) represent the maxima of SNR measured in the co- and cross-channels, respectively. The black boxes show where the presence of supercooled liquid droplets was identified while the red boxes highlight the presence of multiple co-located hydrometeor populations.





a sudden increase of the Doppler velocity of the main spectral mode from $v \approx -0.8\,\mathrm{m\,s^{-1}}$ to $v \approx -1.5\,\mathrm{m\,s^{-1}}$ occurs. This

increase in fall velocity of the main ice crystal population is likely triggered by the riming process highlighted previously. At altitudes below $1.8\,\mathrm{km}$ height, the temperature ranged from $-10°\mathrm{C}$ to $-5°\mathrm{C}$, which is associated to favourable conditions for the riming process as observed in this case. Below $1\,\mathrm{km}$ height, three secondary spectral modes coexist, as highlighted by the red rectangles in all sub-panels of Fig. 10. In Fig. 10a, one dedicated spectral mode is visible at Doppler velocities $v \approx 0\,\mathrm{m\,s^{-1}}$. Low values of $\mathrm{SNR_{co}}$ of this mode and the absence of any associated values of $\mathrm{SNR_{cross}}$ and SLDR (Fig. 10b

and 10c, respectively) indicates the presence of drizzle. Co-located to the drizzle mode, another mode is visible at Doppler velocities from $v \approx -1\,\mathrm{m\,s^{-1}}$ to $v \approx 0\,\mathrm{m\,s^{-1}}$. It is associated to increased values of $\mathrm{SNR_{co}}$ and $\mathrm{SNR_{cross}}$, and high values of SLDR (Figs. 10a, 10b, and 10c, respectively). Its co-location to the drizzle mode highlights that prolate ice particles defined by $\xi > 1$ (Fig. 10e) are produced under the presence of supercooled liquid droplets at temperature of around $-5°\mathrm{C}$. Finally, particles falling at a Doppler velocity $v < -1\,\mathrm{m\,s^{-1}}$ represent the main ice crystal population in the cloud. The Hallett-Mossop

process can be responsible in this case for secondary ice production (SIP) which leads to the formation of columnar crystals, as confirmed by $\xi > 1$ which were derived with the VDPS method using the main peak of $\mathrm{SNR_{cross}}$ (Fig. 9e). Simultaneously, partially rimed aggregates of polarizability ratio of $\xi \approx 1$ are identified and characterized by the VDPS method, based on the main peak of $\mathrm{SNR_{co}}$, similar to what was reported by Billault-Roux et al. (2023). However, another plausible explanation for this secondary ice production of columnar crystals can be due to a fragmentation of freezing drizzle droplets, which was

reported to occur at low temperatures and in the presence of co-located ice crystals such as graupel and supercooled liquid droplets such as drizzle (Luke et al., 2021).

As shown in Fig. 9c, columnar crystals and rimed aggregates were detected by VISSS at 22:08:30 UTC. In contrast only columnar crystals were observed at 22:09 UTC. The disappearance of rimed aggregates in the VISSS measurements can be explained by the increasing dominance of columnar crystals over rimed aggregates. This is reflected in Fig. 10b where $\mathrm{SNR_{cross}}$

of the columnar branch ($v \approx -1\,\mathrm{m\,s^{-1}}$) increases more strongly than in the rimed aggregates branch ($v \approx -2\,\mathrm{m\,s^{-1}}$). The SLDR calculation based on the SNRs observed at the location of the main peak of $\mathrm{SNR_{co}}$ can influence the particle shape inferred by the VDPS method in cases where multiple ice crystal types coexist and the dominant hydrometeor population is shifting. In conclusion, in this case study the VDPS method is able to identify the co-located presence of rimed aggregates as isometric particles defined by a polarizablitiy ratio $\xi \approx 1$, and needles as prolate particles with a polarizability ratio $\xi > 1.2$

using the novel co-cross SLDR calculation procedure.

## 5 Assessment of the impact of non-Rayleigh scattering on the VDPS method

This section aims to quantify the influence of the non-Rayleigh scattering on the performance of the VDPS method. Since the VDPS approach relies on a scattering model based on the Rayleigh approximation (see Sec. 3.1), deviations from the Rayleigh regime can potentially introduce biases and impact the inferred vertical distribution of the polarizability ratio. Non-Rayleigh

scattering occurs when hydrometeor sizes are comparable to the radar wavelength or larger, causing complex scattering behavior that deviate from the simpler Rayleigh approximation. This effect becomes significant in radar systems especially at




shorter wavelengths like Ka- and W-bands ($8.55\,\mathrm{mm}$ and $3.19\,\mathrm{mm}$, respectively) and can impact the interpretation of radar measurements such as reflectivity and polarimetric variables. In a first step, time periods were selected when the worst case, i.e. occurrence of large and dense particles in the VISSS observations and of high values of $\mathrm{DWR_{Ka-W}}$. For these periods, the vertical distribution of the polarizablity ratio delivered by the VDPS method is derived in order to evaluate the impact of the non-Rayleigh scattering regime on polarimetric variables such as SLDR and on the viability of the VDPS method to identify the shape regime of large and/or dense ice particles. The cross-correlation coefficient $\rho_{cx}$ introduced in Sec. 2.2 is used to support the advanced interpretation. In the following subsections, two case studies of large aggregates and quasi melted graupel particles will be evaluated, respectively.

## 5.1 Large aggregates observed on 9 January 2024, between 17:50 and 18:30 UTC

The first case study focuses on the period from 17:00 to 19:00 UTC on 9 January 2024, during which aggregates formed in the columnar crystal growth regime were observed. An overview about the scenario is provided in Fig. 11. In the course of the observation period, the cloud-top temperature of the precipitation system was constantly below $-20°\mathrm{C}$. As the discussion of the case concentrates on the lower parts of the cloud system, the visualization is restricted to the height range from 0 to $4\,\mathrm{km}$ above the ground. The time period of interest ranges from 17:30 to 18:30 UTC, when a fallstreak of high-reflectivity (Fig. 11a), elevated DWR (Fig. 11b) and pronounced LDR signatures especially at W-band (Fig. 11c) and partially at Ka-band (Fig. 11d) was observed at heights between $2.8\,\mathrm{km}$ and the surface. According to VISSS measurements shown in Fig. 11e, large particles are detected at the surface between 17:50 and 18:30 UTC, where the diameter of hydrometeors reaches up to $15\,\mathrm{mm}$. Such values are no longer compatible with the Rayleigh scattering regime for both Ka- and W-band radars. Also the high values of $\mathrm{DWR_{Ka-W}}$ during this time period indicate the presence of large particles causing the W-band to depart from the Rayleigh scattering regime. The formation of the large particles begins at around 2.8 km height where $\mathrm{DWR_{Ka-W}}$ ranges from 6 to 8 dB. At this altitude, the temperature is between $-15$ to $-10°\mathrm{C}$ (Fig. 11) where aggregation is known to be rather efficient (Field and Heymsfield, 2003). This zone represented by the black rectangle in Fig. 11b, is correlated with high values of LDR from RPG94 (Fig. 11c). A decrease in $\mathrm{DWR_{Ka-W}}$ is visible at around 18:08 UTC and coincides with an increase in LDR values from MIRA-35 MBR7 (Fig. 11d), as well as with the presence of larger particles with a diameter exceeding 15 mm (Fig. 11e). The decrease in $\mathrm{DWR_{Ka-W}}$ can indicate a simultaneous departure of the measurements of RPG94 and MIRA-35 MBR7 from the Rayleigh scattering regime, as both bands (Ka- and W-bands) begin to show non-Rayleigh behavior due to the increased particle size and complexity of scattering mechanisms. Indeed, when both bands exit the Rayleigh scattering regime, the differences in backscatter between Ka- and W-band become less pronounced, causing the $\mathrm{DWR_{Ka-W}}$ to decrease or stabilize at a lower level (at around 5 dB in this case).

According to the RHI scans of SLDR and $\rho_{cx}$ presented in Fig. 12a and 12b, respectively, higher values of SLDR (at around $-23\,\mathrm{dB}$) and low values of $\rho_{cx}$ (at around zero) at all elevation angles are measured from 1 km height to the ground. As shown in Fig. 12c, the VDPS method derives slightly prolate particles, exhibiting a polarizability ratio $\xi > 1$. Low values of $\rho_{cx}$ are defined as fully chaotic orientation which can be considered as a special case of reflection symmetry (Myagkov





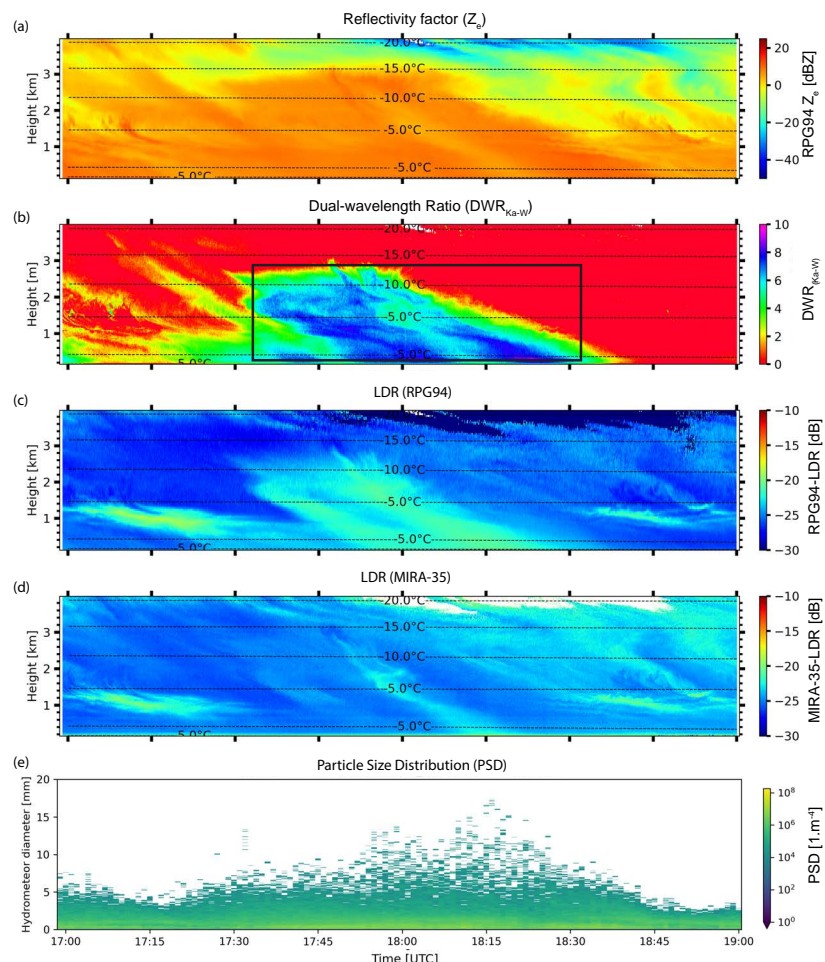

**Figure 11.** Case study of a deep mixed-phase cloud event observed with multi-wavelength polarimetric cloud radars at Eriswil, Switzerland on 9 January 2024 from 17:00 UTC to 19:00 UTC. Vertically pointing W-band (94 GHz) (a) radar reflectivity factor $Z_e$ and (c) the linear depolarization ratio (LDR), and (d) vertically pointing Ka-band (35 GHz) linear depolarization raio (LDR). (b) The dual-wavelength ratio (DWR$_{Ka-W}$, ratio of $Z_e$ between Ka-band MIRA-35 MBR7 and W-band RPG94) and (e) the temporal evolution of the particle size distribution (PSD) from VISSS measurements. The black rectangle highlights the focus area of the case study.





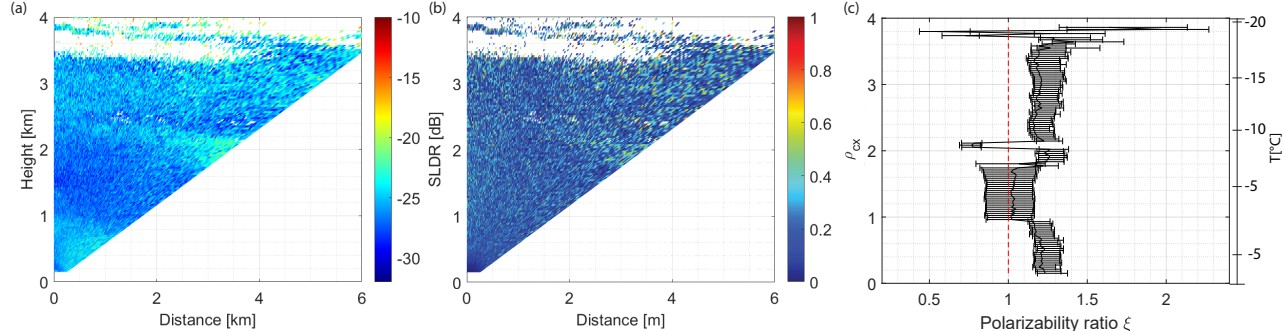

**Figure 12.** RHI scans of (a) SLDR and (b) $\rho_{cx}$ from $90°$ to $150°$ elevation angle recorded by MIRA-35 MBR5 on 9 January 2024 from 18:08:14 to 18:10:14 UTC, in Erwisil, Switzerland. (b) shows the vertical distribution of the polarizability ratio $\xi$ derived using the VDPS method and is correlated with the RHI scans presented in (a) and (b), with the temperature range on the right side.

et al., 2016b), and are not compatible with columnar crystals, characterized as prolate particles which are mostly horizontally oriented. However they can be associated with a polarizability ratio $\xi \approx 1$, characterizing isometric particles.

Examining the Doppler spectrogram presented in Fig. 13c, low SLDR values at around $-30$ dB are observed in regions where the particle fall velocity is v $\approx -1$ m s$^{-1}$. As the fall velocity increases to v $= -2$ m s$^{-1}$, the SLDR rises to about $-20$ dB, as highlighted by the black rectangle. This increase in SLDR is not correlated with any secondary ice population in SNR$_{co}$, as

shown in Fig. 13a. This indicates that only one hydrometeor population is present at Doppler velocities between $v = -1$ m s$^{-1}$ and v $= -2$ m s$^{-1}$. However, the high values of SLDR in this region are correlated with low values of SNR highlighting that this hydrometeor population has a low particle concentration, consistent with the low concentration attributed to the large particles detected with VISSS (Diameter$> 10$ mm) shown in Fig. 11e. Large aggregated ice particles that are associated with higher fall velocities, lead to the non-Rayleigh scattering regime for both RPG94 and MIRA-35 as SLDR increases in Figs. 11c

and 11d, respectively. In Fig. 12, values of SLDR are reaching only $-23$ dB in the RHI scan while SLDR maximum values are higher according to the Doppler spectrogram presented in Fig. 13c. This occurs because SLDR is calculated using the main peak of SNR$_{co}$ defining the prominent hydrometeor population contained in the cloud, reducing the margin of error. Indeed, the particles which are falling slowly ($v \approx -1$ m s$^{-1}$) exhibit lower SLDR values at around $-27$ dB describing isometric particles. Finally, on the right side of the Doppler spectrogram where ice particles are falling with $v > -1$ m s$^{-1}$, high values of SLDR

($> -15$ dB) are in general associated with prolate particles such as columnar crystals. During the same time period, VISSS observations (Fig. 14) reveals a high number of aggregates and a low concentration of columnar crystals. A range of aggregate sizes is shown in Fig. 14 corresponding to the spectrum of fall velocities observed in the Doppler spectrograms shown in Fig. 13.

In the case of large aggregates, columns do indeed cluster together in all possible orientations (see Fig. 14). The internal

microphysics and the low-density structure containing small needles can amplify the crystal depolarization and increase the




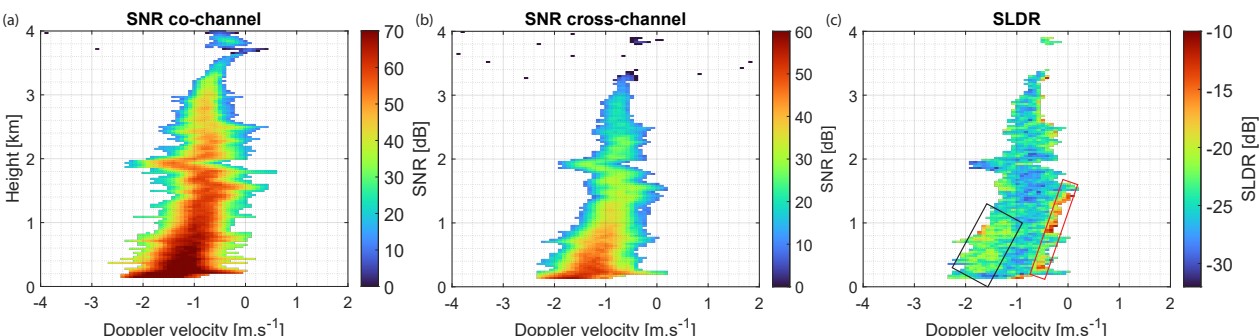

**Figure 13.** Doppler spectrograms at zenith pointing of (a) SNR in the co-channel, (b) SNR in the cross-channel and (c) SLDR measured on 9 January 2024, at 18:08:14 UTC. The black box corresponds to larger aggregates exhibiting higher SLDR values, while the red box highlights columnar crystals characterized by elevated SLDR values.

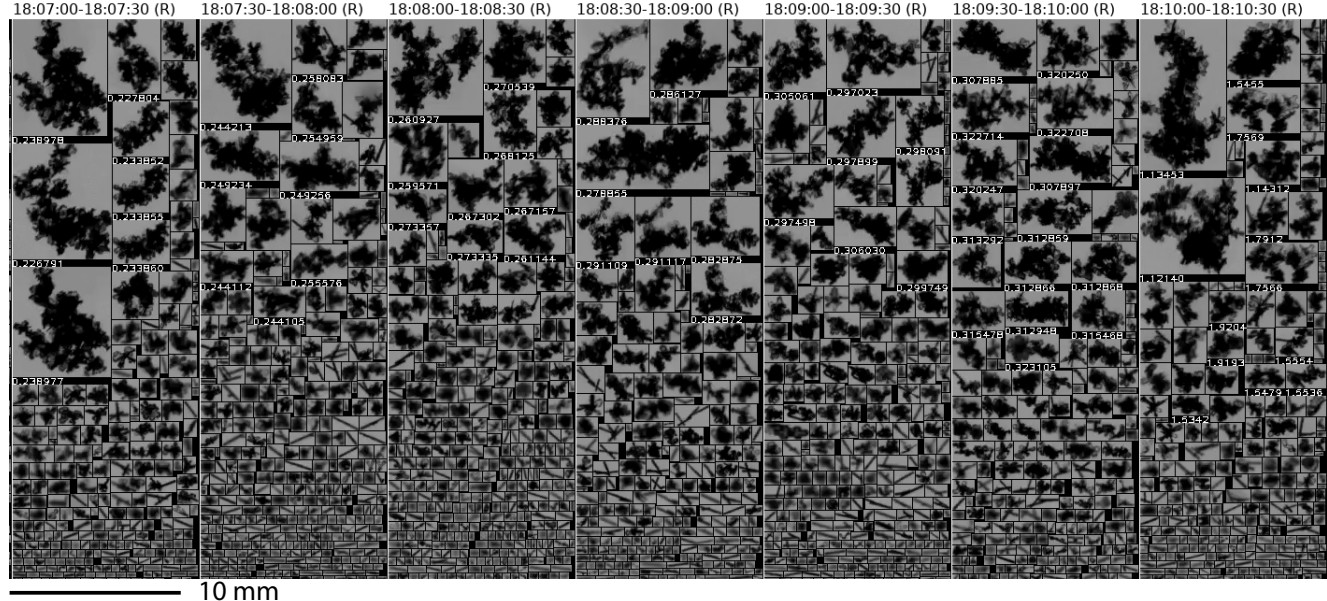

**Figure 14.** Surface observation of hydrometeor shapes detected with VISSS from 18:07:00 to 18:10:30 UTC on 9 January 2024.



SLDR at all elevation angles. The inclusion of $\rho_{cx}$ to the VDPS method under conditions of large ice crystals exhibiting high SLDR values may hinder the detection of large aggregates primarily composed of columnar crystals.

In conclusion, the increase in LDR in the W-band radar observations is intrinsically correlated with the occurrence of higher values of $DWR_{Ka-W}$ (highlighted in the black box in Fig. 11), indicating that the departure from the Rayleigh scattering regime

occurs slightly earlier (in time as aggregation is ongoing) due to the smaller wavelength and influences the LDR. In contrast, LDR values from the Ka-band radar increase only when $DWR_{Ka-W}$ decreases, typically when the ice particle diameter reaches 10–15 mm, allowing a broader range of particle sizes to form without significantly affecting the LDR.

Consequently, the VDPS method is using at this stage the MIRA-35 cloud radars, operating at Ka-band, and the impact of the non-Rayleigh scattering will occur for very large particles, only in this regard, the most of aggregates will be detectable

as isometric particles. In the case of actually isometric aggregates larger then 15 mm, the VDPS method will derive slightly prolate shaped particles with a polarizability ratio ranging from 1.2 to 1.3 which is already considered as quasi isometric particles (Fig. 12c). Nevertheless for any potential future version of the VDPS method using a W-band radar, the non-Rayleigh scattering produced by particles larger than 5 mm has to be taken into account riming and aggregation processes.

### 5.1.1  Graupel observed on 24 February 2024, between 12:50 and 13:10 UTC

The second case study presents large graupel that were detected between 12:50 and 13:10 UTC on 24 February 2024. An overview of the observational period is presented in Fig. 15 in which the actual time and range span of the case study is shown by the black rectangle in Fig. 15b. In this area, high $DWR_{Ka-W}$ values ($DWR_{Ka-W} > 10\,dB$) suggest a departure of RPG94 from the Rayleigh-scattering regime and thus indicate the presence of large particles. At heights below 2 km, the $DWR_{Ka-W}$ at its maximum values and the temperature ranges between $-10°C$ and $5°C$. The model-based $0°C$-isotherm is located at 0.5 km

height which suggests that large graupel particles require additional time to undergo melting from 0.5 km height to the ground and are nearly melted enhancing non-Rayleigh scattering due to resonance and attenuation effects by potential occurrence of liquid water. Additionally, the high density of graupel particles further accelerates the departure of the backscattering properties from the Rayleigh scattering regime. Fig. 15e shows an increasing number of large particles in the size distribution (PSD) with hydrometer diameters reaching 6 mm precisely when $DWR_{Ka-W}$ exhibits highest values.

Despite the W-band radar RPG94 having exited the Rayleigh scattering regime producing high values of $DWR_{Ka-W}$, LDR values measured with both W- and Ka-band radars remain low, as shown in Figs. 15c and 15d, respectively. Regarding the RHI scans presented in Fig. 16 and 16b, low values of SLDR and $\rho_{cx}$ are observed below 1 km in height, and from zenith to low elevation angles, respectively, while the VDPS method derives only isometric particles characterized by a polarizability ratio $\xi \approx 1$, as shown in Fig. 16c. This particle shape is confirmed by VISSS measurements shown in Fig. 18 where it detected

simultaneously large and spherical graupel during the same time period.

Regarding the $SNR_{co}$ Doppler spectrogram presented in Fig. 17a, the coexistence of two hydrometeor populations is revealed below 0.7 km, which corresponds to the altitude where the model temperature reaches $0°C$. Indeed, as graupel begin to melt during their descent, they may undergo mechanical fragmentation due to structural weakening, internal pressure, or aerodynamic stress. This process can generate secondary ice particles or droplets, contributing to a more diverse hydrometeor



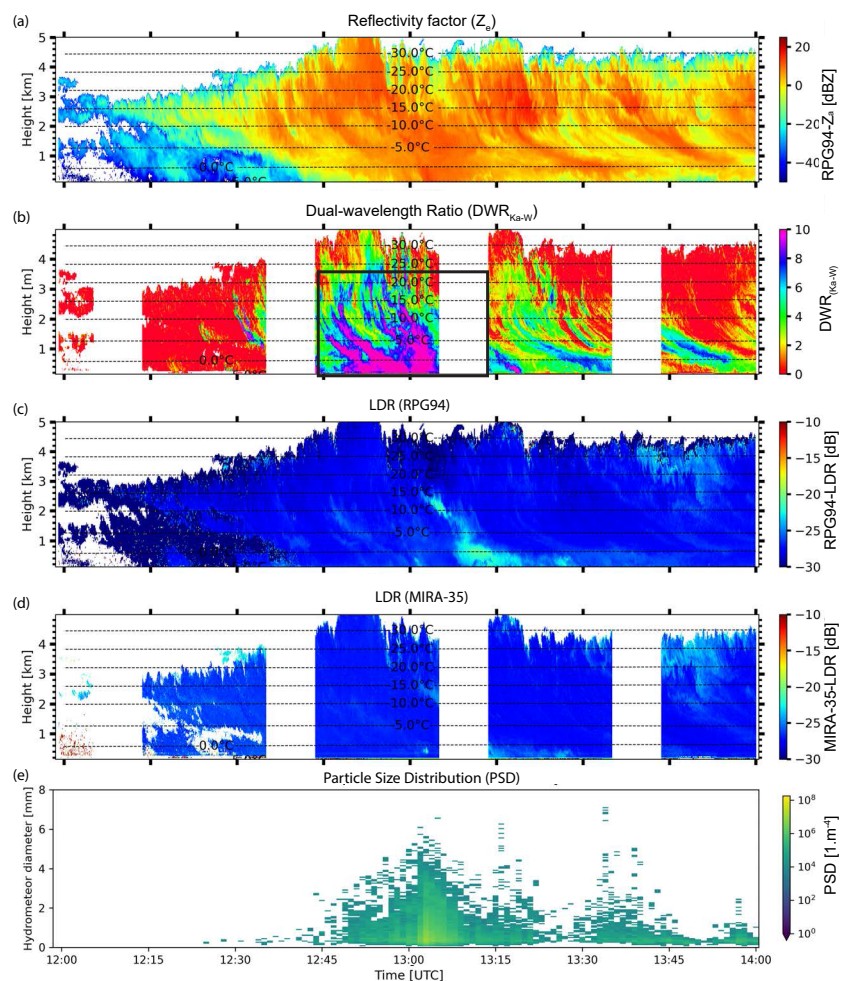

**Figure 15.** Case study of a deep mixed-phase cloud event observed with multi-wavelength polarimetric cloud radars at Eriswil, Switzerland, on 24 February 2024 from 12:00 UTC to 14:00 UTC. Vertically pointing W-band (94 GHz) (a) radar reflectivity factor $Z_e$ and (c) the linear depolarization ratio (LDR), and (d) vertically pointing Ka-band (35 GHz) linear depolarization raio (LDR). (b) The dual-wavelength ratio (DWR$_{Ka-W}$, ratio of $Z_e$ between Ka-band MIRA-35 MBR7 and W-band RPG94) and (e) the temporal evolution of the particle size distribution (PSD) from VISSS measurements. The black rectangle highlights the focus area of the case study. No measurements are available for the time periods where white bands are shown in (b), due to RHI and PPI scans that were conducted during these periods.





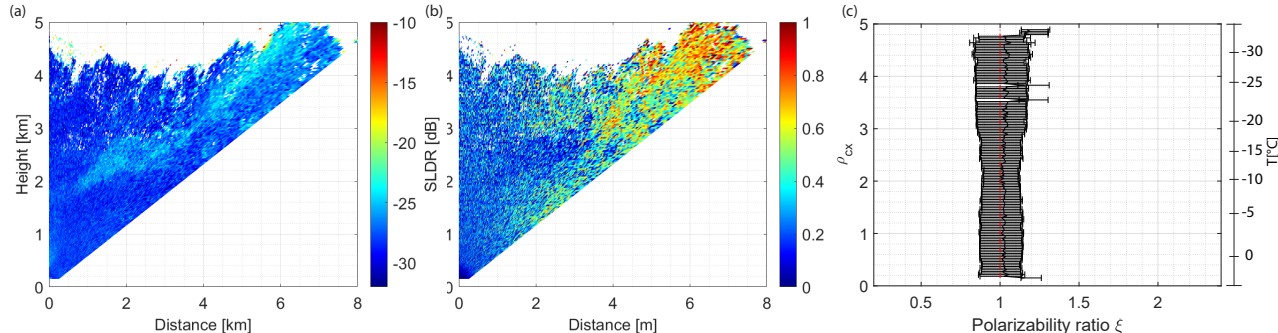

**Figure 16.** RHI-scans of (a) SLDR and (b) $\rho_{\mathrm{cx}}$ from $90°$ to $150°$ elevation angle recorded by MIRA-35 MBR5 on 24 February 2024 from 13:06:05 to 10:08:13 UTC, in Erwisil, Switzerland. (c) shows the vertical distribution of the polarizability ratio $\xi$ derived from the VDPS method and correlated with the RHI scans presented in (a) and (b), with the temperature range on the right side.

population, as shown in Fig. 18 where the VISSS detects a variety of hydrometeor shapes. Such fragmentation is especially likely near the $0°$C isotherm, where a mixture of solid and liquid phases coexists. The radar reflectivity factor $Z_{\mathrm{e}}$ is enhanced and the polarimetric signatures are drastically altered. Moreover, the graupel becomes denser and less irregular, its surface become smoother, and its aerodynamic properties evolve, resulting in an increased fall velocity, consistent with the population falling at $v \approx -4.5 \,\mathrm{m\,s^{-1}}$ shown in Fig. 17a. According to Fig. 17c, the population of dense graupel exhibits low SLDR val-

ues represented in the red box, while the other, more slowly falling ice population ($v \approx -2.5 \,\mathrm{m\,s^{-1}}$, highlighted by the black rectangle) is characterized by higher SLDR values, indicative of smaller, prolate-shaped particles, as it was also seen by the VISSS in Fig. 18. Unlike the previous case study involving needles aggregates, the internal microphysical structure of graupel does not consist of columnar crystals, and the particles tend to appear nearly spherical. This spherical geometry may explain the low SLDR values observed in cases of quasi-melted graupel while $\mathrm{DWR_{Ka-W}}$ exhibits higher values.

To conclude, the VDPS method derives isometric particles when large graupel particles are observed with the VISSS. The large and relatively warm graupel observed between 12:50 and 13:10 UTC is responsible for the high $\mathrm{DWR_{Ka-W}}$ values, due to its deviation from the Rayleigh scattering regime. In the case of large, dense and warm particles, the non-Rayleigh scattering regime is probably affecting both MIRA-35 cloud radars and RPG94, operating at Ka- and W-band, respectively, but does not impact polarimetric parameters such as SLDR measured with both radars. Consequently, the detection of graupel by the VDPS

method remains consistent and reliable.

## 6   Summary and Conclusions

In this study, the VDPS method was for the first time evaluated and validated through comparison with in-situ measurements. Subsequently, the influence of the non-Rayleigh scattering regime on the detection of large aggregates and graupel was assessed. Earlier studies have demonstrated the potential of the polarimetric parameter SLDR to describe the shape of particles

(Matrosov et al., 2012; Myagkov et al., 2016b; Teisseire et al., 2024, 2025). In this article, the VDPS method was contextu-





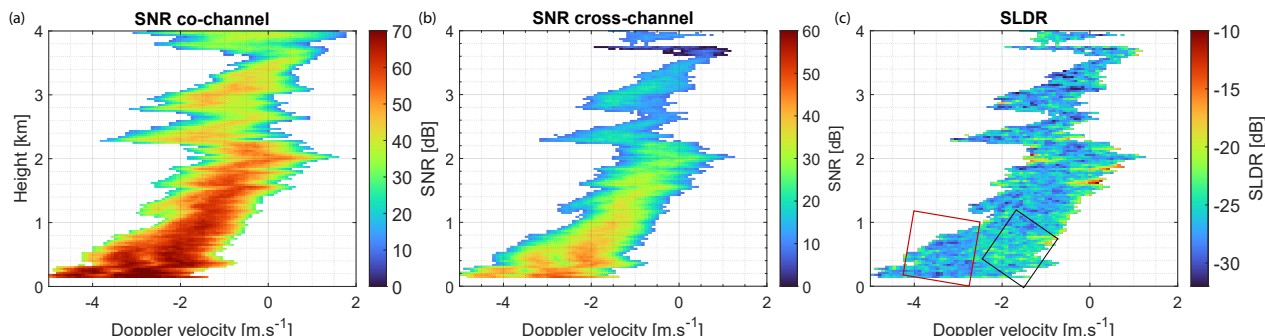

**Figure 17.** Doppler spectrograms at zenith pointing of (a) SNR in the co-channel, (b) SNR in the cross-channel and (c) SLDR measured on 24 February 2024, at 13:08:12 UTC. The black box shows a hydrometeor population which depolarizes more strongly compared to the hydrometeor population shown in the red box.

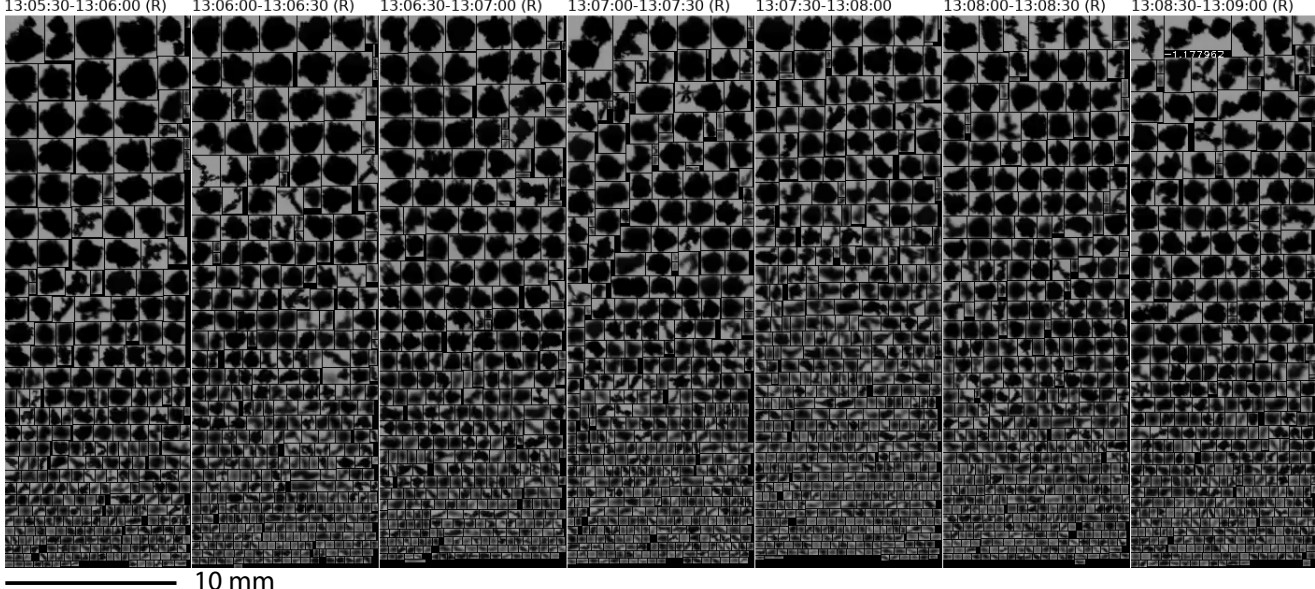

**Figure 18.** The surface observation of hydrometeor shapes detected with VISSS from 13:05:30 to 13:08:30 UTC on 24 February 2024.





alized with VISSS measurements and multi-wavelength spectral cloud radar methods by means of several case studies during which well-defined hydrometeor characteristics, such as the presence of pristine hydrometeors, graupel or aggregates were present. The VDPS method was demonstrated to be able to derive the shape of small hydrometeors and to assign classified particles to either graupel formation or aggregation processes.

The method proves robust under Rayleigh scattering conditions, but its interpretation in the presence of large aggregates requires careful consideration due to the potential influence of non-Rayleigh scattering effects and the complex actual shapes of such particles. The presented version of the VDPS method utilizes the Ka-band SLDR mode radar MIRA-35 MBR5 (see Sec. 2.2), operating at 35 GHz. An advantage of using this radar type is that particles must be relatively large (hydrometeor diameter $> 5$ mm) in order to cause the backscattered signals to depart from the Rayleigh scattering regime, making it suitable

for analyzing even large hydrometeors such as aggregates and graupel.

In a first case study involving large needle-based aggregates detected by the VISSS, an increase in SLDR was observed with the W-band radar RPG94, coinciding with high $DWR_{Ka-W}$ values that indicate a departure from the Rayleigh scattering regime for the W-band. An increase in SLDR with the Ka-band radar is observed only when DWR values begin to decrease slightly, which may indicate a simultaneous departure from the Rayleigh scattering regime for both W- and Ka-band radars.

Unfortunately, no X-band radar data were available for this date to confirm the departure of the Ka- and W-band cloud radars from the Rayleigh regime. As the X-band radar operates at a wavelength of 3 cm, it may not yet be significantly affected by non-Rayleigh scattering. From the analysis, it was found that non-Rayleigh scattering leads to an increase of the SLDR at all elevation angles of the RHI scans underlying the VDPS method. This results in the introduction of a positive bias in the retrieved polarizability ratio $\xi$, leading to a tendency of the VDPS method to detect prolate particles when actually isometric

particles are observed. The margin of error remains low, with the primary risk being a missed aggregation detection due to the misclassification of aggregates as prolate-shaped particles. As an additional measure to identify the bias in $\xi$ by non-Rayleigh scattering, the cross-correlation coefficient $\rho_{cx}$ can be utilized. In the actual presence of prolate hydrometeors $\rho_{cx}$ should increase with increasing off-zenith angle of the RHI scan. When isometric particles are observed, $\rho_{cx}$ is constantly low at all elevation angles. Thus, while the VDPS method relies solely on SLDR to infer particle shape, $\rho_{cx}$ was found to support the

identification of the presence of non-Rayleigh-induced prolate signatures.

Motivated by the VISSS observations of the co-existence of multiple hydrometeor types in the same volume, an extension of the VDPS method was introduced in the framework of this study. The original retrieval was based solely on the SLDR calculated for the location in the Doppler spectra where the main peak of the SNR in the co-channel was observed. The newly introduced extended approach consists of performing the calculation for the part of the Doppler spectrum where the main peak

in the SNR of the cross-channel was observed. This ultimately enables one to identify the overall dominating hydrometeor type from co-channel peak signal, while the approach that used the position of the main peak in the cross-channel SNR enables the identification of the polarimetrically dominating hydrometeor type, such as secondary prolate-shaped ice.



*Code and data availability.* The cloud radar raw data, temperature data, RHI scan data and VISSS measurements are available at https://doi.org/10.5281/zenodo.15692821.15692821. The Cloudnet datasets are provided by the ACTRIS Data Centre node for cloud profiling via the following links: https://hdl.handle.net/21.12132/2.b6c194d7d33b448e. For plotting of the data, the tool pyLARDA was used, which is available at https://doi.org/10.5281/zenodo.4721311 (Bühl et al., 2021), and the VISSS data at https://doi.pangaea.de/10.1594/PANGAEA.981222 (Maahn and Ettrichrätz, 2025). The VDPS algorithm is available upon request.

*Author contributions.* AT developed the VDPS method, analysed the data and drafted the manuscript. PS, KO, RS and JH supervised the CLOUDLAB campaign. MM has deployed the VISSS instrument and stored the data . All authors contributed to edit and review the manuscript.

*Competing interests.* At least one of the (co-)authors is a member of the editorial board of Atmospheric Measurement Techniques.

*Financial support.* Funding for this study was provided by the Deutsche Forschungsgemeinschaft (DFG, German Research Foundation) within the priority program SPP 2115 PROM via project numbers 408008112 (PICNICC, CORSIPP) and 408027490 (PolarCAP, SPOMC), by the European Union's Horizon Europe projects CleanCloud (grant no. 101137639), and the European Union's Horizon 2020 research and innovation program (CLOUDLAB, grant agreement no. 101021272). The LACROS infrastructure received financial support via ACTRIS-D, which is funded by the Federal Ministry of Education and Research of Germany under the funding code 01LK2001A.

*Acknowledgements.* We gratefully acknowledge the ACTRIS Cloud Remote Sensing Unit for making the Cloudnet datasets publicly available. We also thank Martin Radenz and Andi Klamt for their support with PyLARDA, and Anne-Claire Billault-Roux for her valuable guidance on interpreting Doppler spectrograms. We are grateful to Leonie von Terzi and Dmitri Moisseev for their insights regarding dual-frequency radar measurements and the non-Rayleigh scattering regime.



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
