# Peer review of "Evaluation of the Vertical Distribution of Particle Shape (VDPS) method with in situ measurements and assessment of the impact of non-Rayleigh scattering"

_EGUsphere, 2025_

## Referee Comment (RC1)

**Review of Evaluation of the VDPS method with in-situ measurements and assessment of the impact of non-Rayleigh scattering**

General comment:
The study evaluated the VPDS method with the VISSS instrument during the CLOUDLAB campaign. It is shown that the retrieved polarizability ratio and consequent interpretation of the particle type are consistent with the on-ground observations of the VISSS. The study further tries to evaluate the Rayleigh assumption that is used in the VDPS method, concluding that for large aggregates, at Ka or W-Band deviations can be expected. I think this study is a nice addition to the series of papers on the VDPS method, as for the first time the method is evaluated. I however have several suggestions that would make the evaluation of the Rayleigh assumption stronger, especially a discussion about non-Rayleigh scattering from ice crystals at non-zenith observations (see comment 6) has to be added in, if you want to evaluate the Rayleigh assumption correctly.

**Major comments:**

1. If you want to evaluate non-Rayleigh scattering effect wouldn't it be best if you simulate SLDR with existing scattering databases such as the Lu (2016) database? They have aggregates and ice crystals, since you are looking at the main SNR peak, both particle types are possible. Forward simulate SLDR for the different particle types at different radar elevation angles and see if non-Rayleigh effects are visible (i.e. dips, differences between X-Band, Ka-Band and W-Band). If you know that the variable you use for retrieving the shape has Rayleigh-scattering effects, then probably also your retrieval is influenced by it. In any case, I would be very interested to see how SLDR varies with increasing sizes and particle types, as I have thus far only seen similar calculations for ZDR.

2. You are calculating DWR integrated over the entire spectrum, and then compare it to the retrieved shape of the particles populating the main peak. Usually the largest particles drive DWR, which might not be the particles populating the main peak. I would suggest calculating the DWR of the main peak, and to not use the variable integrated over the entire spectrum, because otherwise you are likely comparing two different populations.

3. I would also suggest looking at spectral DWR, find the regions of high DWR and look at the corresponding LDR (SLDR) spectrum in this region to see how SLDR looks like there.

4. Your comparison against in-situ seems to be done "visually", is there a possibility to analyse it more objectively?

5. In my opinion more information from the radars and other instruments can be used to make your discussion of e.g. riming stronger. The mean Doppler velocity is a variable used in most studies to show riming, as riming strongly increases the fall velocity of ice particles. Also, I find it very hard to actually determine super-cooled liquid droplets in the Doppler spectra you showed. Just because there is a signal around 0 m/s does not mean that these are droplets, to me it rather looks like your spectrum gets shifted there due to updrafts. In your Figure 5d, also the region above your rectangle has the same criteria as you described in your text, why did you not classify this as droplets then? Is there a LIDAR or ceilometer available at the side? Since this is a snowfall case it would be easy to use the LIDAR to detect the height at which droplets are

expected. If this is not possible then I would suggest to use values from literature that define where it is likely to have a droplet peak in the spectrum (i.e. Ze smaller than a certain threshold)

6. I am missing a discussion on non-Rayleigh scattering of crystals at non-zenith viewing angles. The scattering properties of the particles depends strongly on the viewing angle, as these particles are typically highly asymmetric. When viewed zenith, the electromagnetic (EM) wave only travels through a small amount of mass, as the particles are oriented with their largest dimension approximately horizontal. However, when viewed from the side, the mass (size) the EM wave has to travel through is much larger, thus causing non-Rayleigh effects also for ice crystals. This effect can be observed when DWR from non-zenith observations is analysed. In your discussion you are only looking for regions where DWR is large when looking zenith, thus only looking at cases where aggregates can cause non-Rayleigh scattering. However, your method uses different elevation angles, thus also having ice crystals that are non-Rayleigh scatterers. You could investigate this behaviour by calculating LDR (SLDR) of ice crystals at different elevation angles as described in comment 1.

**Other comments:**

How well do you trust your temperature information from IFS? Especially in case of an inversion, IFS is known to not represent the temperature well. I don't know how well IFS performs in mountain areas. While the temperature information is not that essential for your paper, you do use it to discuss the shape retrieval and possible microphysical processes that happen, so in my opinion it would be worth to check if the temperature from IFS is performing reasonably well.

Line 30-31: the fall velocity of ice crystals is not necessarily smaller than -1m/s (see Karrer et al. 2020), The fall velocity depends on the size, and for larger sizes particles such as plates or columns actually fall much faster than aggregates, whose fall velocity saturates at -1m/s.

Line 117: you are saying you use a spheroidal scattering model? I thought the method is based on Rayleigh right? Spheroidal can mean many things (e.g. T-matrix, Mie,..), I would be more specific here.

Line 126 (and other places): you are saying that the peak of your spectrum represents the dominant hydrometeor population, however, how can you be sure it is only one hydrometeor type in this Doppler bin? From my understanding there is no clear way to determine that there is not a mixture of particles present in each Doppler bin.

Table 1: you write frequency and wavelength in row 1, however, you only mention the frequency in the following rows.

Line 171: you say ice crystal population, I would rather refer to it as ice particle population, because you do not know what particle type it is.

Figure 1: I found it hard to compare DWR to Ze and LDR, because of the different colormap used, where red in DWR means small, but red in Ze means large. For simplicity I would suggest to use the same colormap for all variables.

Line 198-200 and line 234: where are the dendrites coming from? Dendrites usually form at -15°C, at -10°C it is typically assumed that plates are forming, not dendrites.

Figure 3: the -10°C is displayed twice, is this correct?

Paragraph from Line 337 to 344: I did not understand everything here, perhaps you could explain it differently?

Paragraph from Line 355 to 365: I am not sure I agree here… DWR KaW should be stable at larger values than 5dB, especially if you reached 8dB before. The drop is not 3dB, when both are in the non-Rayleigh regime, but rather 1-2dB. Are you sure that your PSD stays the same in the analysed fall streak? Because DWR is also quite strongly dependent on the shape of your PSD…

Line 375: you say that because you don't see any clear separation of hydrometeor species in the Doppler spectra (I am assuming you mean any new peak appearing), there is only this one present. I don't think you can say that easily, just because there is no new peak doesn't mean that it is the same particle population in these bins.

Literature:
Lu, Yinghui, et al. "A polarimetric scattering database for non-spherical ice particles at microwave wavelengths." *Atmospheric Measurement Techniques* 9.10 (2016): 5119-5134.